# FEW-SHOT TRANSFERABLE ROBUST REPRESENTATION LEARNING VIA BILEVEL ATTACKS

## ABSTRACT

Existing adversarial learning methods for enhancing the robustness of deep neural networks assume the availability of a large amount of data from which we can generate adversarial examples. However, in an adversarial meta-learning setting, the model needs to train with only a few adversarial examples to learn a robust model for unseen tasks, which is a very difficult goal to achieve. Further, learning transferable robust representations for unseen domains is a difficult problem even with a large amount of data. To tackle such a challenge, we propose a novel adversarial self-supervised meta-learning framework with bilevel attacks which aims to learn robust representations that can generalize across tasks and domains. Specifically, in the inner loop, we update the parameters of the given encoder by taking inner gradient steps using two different sets of augmented samples, and generate adversarial examples for each view by maximizing the instance classification loss. Then, in the outer loop, we meta-learn the encoder parameter to maximize the agreement between the two adversarial examples, which enables it to learn robust representations. We experimentally validate the effectiveness of our approach on unseen domain adaptation tasks, on which it achieves impressive performance. Specifically, our method significantly outperforms the state-of-the-art meta-adversarial learning methods on few-shot learning tasks, as well as self-supervised learning baselines in standard learning settings with large-scale datasets.

## 1 INTRODUCTION

Deep neural networks (DNNs) are known to be vulnerable to imperceptible small perturbations in the input data instances (Szegedy et al., 2013). To overcome such adversarial vulnerability of DNNs, adversarial training (AT) (Madry et al., 2018), which trains the model with adversarially perturbed training examples, has been extensively studied to enhance the robustness of the trained deep network models. While the vast majority of previous studies (Zhang et al., 2019; Carlini & Wagner, 2017; Moosavi-Dezfooli et al., 2016; Wang et al., 2019; Rebuffi et al., 2021) have been proposed to defend against the adversarial attacks that maximize the classification loss, they assume the availability of a large amount of labeled data. Even with the recent progress in adversarial supervised learning, training on a large number of samples is essential to achieve better robustness (Carmon et al., 2019; Rebuffi et al., 2021; Gowal et al., 2021). Recently, Carmon et al. (2019) employs larger dataset (i.e., TinyImageNet (Le & Yang, 2015)) with pseudo labels, Gowal et al. (2021) utilizes generative model to generate additional samples from the dataset, and Rebuffi et al. (2021) leverages augmentation functions to obtain more data samples.

On the other hand, a meta-learning framework (Koch et al., 2015; Sung et al., 2018; Snell et al., 2017; Finn et al., 2017; Nichol et al., 2018) which learns to adapt to a new task quickly with only a small amount of data, has been also known to be vulnerable to adversarial attacks (Goldblum et al., 2020). Since meta-learning employs scarce data and has to adapt quickly to new tasks, it is difficult to obtain robustness with conventional adversarial training methods which require a large amount of data (Goldblum et al., 2020). Adversarial Querying (AQ) (Goldblum et al., 2020) proposed an adversarially robust meta-learning scheme that meta-learns with adversarial perturbed query examples with AT loss (Madry et al., 2018). Similarly, Wang et al. (2021) studies how to enhance the robustness of a meta-learning framework with the adversarial regularizer in the inner adaption or outer optimization. However, previous works (Goldblum et al., 2020; Wang et al., 2021) show poor robustness on unseen domains (see Table 1).

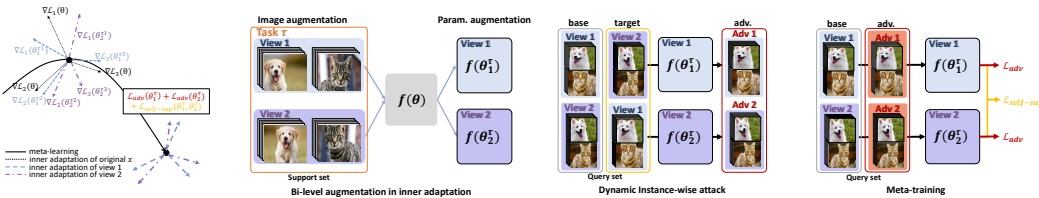

(a) Robust meta-learning      (b) Inner adaptation      (c) Meta optimization

Figure 1: **Overview of TROBA.** (a) TROBA adapts the encoder to differently augmented sets of the support sets (blue, purple line). Then, it meta learns (black line) with both adversarial loss (red) and self-supervised loss (yellow). (b) During the inner adaptation, TROBA adapts encoders with the differently augmented support sets. (c) To generate adversarial examples for meta-learning, we propose a bilevel attack with the instance-wise attack that maximizes the difference between differently augmented query images, for the task-shared encoder $f$. Then, we train the framework to have an adversarially consistent prediction across multiple views with self-supervised loss while learning the encoder to generalize across tasks, which enables it to learn robust representations that are transferable to unseen tasks and domains.

Since existing adversarial meta-learning approaches (Yin et al., 2018; Goldblum et al., 2020; Wang et al., 2021) mostly focus on the rapid adaptation to new tasks, while mostly reusing the features with little modification at the task adaptation step (Oh et al., 2020), the representations themselves may not be effectively meta-learned to be robust across tasks, and thus they fail to achieve robustness when applied to unseen datasets (Section 4.1). The t-sne visualization of the feature space of AQ in Figure 2 shows that the embeddings of the adversarial examples have large overlaps across classes, which confirms this point.

To tackle such challenges, we propose a novel and effective adversarial meta-learning framework which can generalize to unseen domains, *Transferable RObust meta-learning via Bilevel Attack (TROBA)*. TROBA utilizes a bilevel attack scheme to meta-learn robust representations that can generalize across tasks and domains, motivated by self-supervised learning (Figure 1). Specifically, we redesign the instance-wise attack proposed in Kim et al. (2020); Jiang et al. (2020) which maximizes the instance classification loss, by adapting the shared encoder to two sets of differently augmented samples of the same instance with inner gradient update steps and then attacking them (dynamic instance-wise attack). Then, our framework learns to maximize the similarity between the feature embeddings of those two attacked samples, while meta-learning the shared encoder by BOIL (Oh et al., 2020), which allows it to learn robust representations for any given set of augmented samples. Since the robustness is achieved at the representation level without consideration of the labels rather than at the task level, our framework can generalize to unseen tasks and domains.

The experimental results from multiple benchmark datasets show that our meta-adversarial learning framework is not only robust on few-shot learning tasks from seen domains (Table 3) but also on tasks from unseen domains (Table 1, 2) thanks to its ability to learn generalizable robust representations. Moreover, our model even obtains comparable robust transferability to the self-supervised pre-trained models while using fewer data instances (Table 7). Our contributions can be summarized as follows:

- We propose a **novel adversarial meta-learning framework** with **bilevel attacks**, which allows the model to learn generalizable robust representations across tasks and domains.
- Our framework obtains impressive robustness in few-shot tasks both in the seen domain and the unseen domains. Notably, on unseen domains, our model outperforms baselines by **more than 10%** in robust accuracy without compromising the clean accuracy.
- Our framework achieves impressive robust transferability in unseen domains, that are **competitive** with that of the model pre-trained by SSL with larger data, while using significantly smaller amount of data for training.

## 2   RELATED WORK

**Meta-Learning.**   Meta-learning (Thrun & Pratt, 1998) aims to learn general knowledge across the distribution of tasks that can be utilized to rapidly adapt to new tasks with a small amount of data. Meta-learning approaches can be broadly categorized into metric-based (Koch et al., 2015; Sung et al., 2018; Snell et al., 2017) or gradient-based (Finn et al., 2017; Nichol et al., 2018) approaches,

and in this work, we focus on the gradient-based approaches to benefit from their versatility. Perhaps the most popular work in this direction is Model Agnostic Meta-Learning (MAML) (Finn et al., 2017) which uses a bilevel optimization scheme. MAML consists of inner- and outer-optimization loops, where in the inner loop, the model takes a few gradient steps to quickly adapt to a new task and in the outer loop, the parameters are meta-updated to generalize multiple tasks. To efficiently reuse the features of the encoder, ANIL (Raghu et al., 2019) updates only the classification head in the inner loop while keeping the encoder fixed. On the other hand, Oh et al. (2020) propose BOIL, which meta-learns the feature extractor while keeping the final classifier fixed, and show that it has better generalization over cross-domain adaption tasks compared to MAML or ANIL. Various approaches have been proposed to efficiently update the meta-learner; Li et al. (2017) propose to meta-learn the learning rate for each parameter, Li et al. (2017) propose to reduce the computation costs of MAML with first-order approximation in the inner loop, and Nichol et al. (2018) propose to repeatedly sample a task to move the initialization toward the task.

**Adversarial Training.** Many existing works aim to enhance the robustness of a model trained with supervised learning with labeled data, by utilizing adversarial examples generated by applying perturbations that can maximize its loss (Goodfellow et al., 2015; Carlini & Wagner, 2017; Papernot et al., 2016). The most popular approach to enhance the robustenss of general DNNs is Adversarial Training from Madry et al. (2018), which utilizes project gradient descent (PGD) to maximize the loss in the inner-maximization loops while minimizing the overall loss on adversarial samples generated by the PGD attack. Zhang et al. (2019) theoretically shows the trade-off between clean accuracy and robustness in adversarial training (TRADES), and introduces regularized Kullback-Leibler divergence (KLD) loss that helps to enhance the robustness by enforcing the consistency in the predictive distribution between the clean and adversarial examples.

**Adversarial Meta-Learning.** Although meta-learning contributes to learning useful generalizable knowledge with scarce data, existing state-of-the-art meta-learning approaches are prone to adversarial perturbations. To tackle this problem, Yin et al. (2018) attempt to combine adversarial training (AT) (Madry et al., 2018) with MAML (Finn et al., 2017), referring to the problem as adversarial meta-learning (ADML). ADML uses clean and adversarial examples simultaneously to update the parameters both in the inner and outer optimization, and thus requires a high computational cost. However, Goldblum et al. (2020) later point out that ADML (Yin et al., 2018) may not obtain good robustness to strong attack since it uses relatively weak attacks during training. Then, they propose an adversarially robust meta-learner, Adversarial Querying (AQ), which trains with adversarial examples only from the query set. Wang et al. (2021) studies how to obtain robustness in meta-learning framework and suggest Robust-regularized meta-learner on top of the MAML (RMAML), where adversarial attacks are conducted only in the meta-optimization phase as in AQ (Goldblum et al., 2020). However, previous adversarial meta-learning methods are still vulnerable to adversarial attacks on unseen domains. This is mainly because they do not result in learning robust representations, as observed in Oh et al. (2020), Goldblum et al. (2020) and Wang et al. (2021), as the representations are reused with little updates during inner optimization steps. Thus the effect of meta-learning is minimal on the learned representations, which prevents them to achieve generalizable robustness. To tackle such a limitation, we propose a robust self-supervised meta-learning framework via bilevel attacks which meta-learns the representation layers to generalize across any adversarial learning tasks that are generated from randomly sampled instances.

## 3 TRANSFERABLE ROBUST META-LEARNING VIA BILEVEL ATTACKS

One of the ultimate goals of meta-learning is the generalization to unseen domains and tasks. Learning transferable robust representations that can generalize to unseen domains is a difficult problem even with a large amount of labeled data. Yet, we aim to tackle this challenging problem only by learning with a few samples per category, but with an effective meta-learning framework with a bilevel attack scheme. Before describing our methods, we first elaborate on the preliminaries of our framework.

### 3.1 PRELIMINARIES

**Model-Agnostic Meta-Learning.** Let us denote the encoder as $f_\theta$ and classifier as $h_\mu$. Since meta-learning aims to learn to learn the new tasks, it needs to train on a large number of tasks $\tau$

sampled from a task distribution $p(\tau)$, where a given task consists of the support set $\mathbf{S}_\tau$ and the query set $\mathbf{Q}_\tau \in \mathbf{D}$. Each set contains a n-way k-shot classification task, that classify n classes with k images, i.e., $n \times k$ instances. The most popular framework for meta-learning is model-agnostic meta-learning (MAML) Finn et al. (2017), which meta-learns the model with a bilevel optimization scheme, with inner optimization and outer meta-level optimization steps. During the inner optimization, we adapt the shared initial parameter to each new task $\tau$ to obtain task-adaptive parameters $\theta^\tau$ and $\mu^\tau$, by taking a few gradient steps, as follows:

$$\theta^\tau, \mu^\tau = \theta, \mu - \alpha \nabla_{\theta,\mu} \mathcal{L}_{\mathbf{S}_\tau}(h_\mu(f_\theta)), \tag{1}$$

where $\mathbf{S}_\tau$ is a support set of task $\tau$, $\alpha$ is the step size and $\mathcal{L}$ is a task-specific loss to conduct gradient step for inner updates (e.g. cross-entropy loss). There also exist different variants of the MAML framework with respect to which parameters to update. ANIL (Raghu et al., 2019) only meta-learns the final linear layer while fixing the encoder (i.e., $\theta^\tau = \theta$), for rapid adaptation to a new task while reusing the features. On the other hand, BOIL (Oh et al., 2020) only meta-learns the encoder, thus the representation layers, while keeping the final classifier fixed (i.e., $\mu^\tau = \mu$). We employ BOIL (Oh et al., 2020) which only updates the encoder because our focus is on learning generalizable robust representations. In the meta-optimization phase, model parameters are updated with meta-objective via stochastic gradient descent (SGD) as follows:

$$\theta, \mu \leftarrow \theta, \mu - \beta \nabla_{\theta,\mu} \sum \mathcal{L}_{\mathbf{Q}_\tau}(h_{\mu^\tau}(f_{\theta^\tau})), \tag{2}$$

where $\mathbf{Q}_\tau$ is a query set of task $\tau$, $\beta$ is a meta step size and $\mathcal{L}$ is meta-objective. The meta-objective is a summation of losses from the query set of all given tasks, where the losses depend on what aims to be meta-learned. To reduce the computation overhead in MAML, we use Meta-SGD (Li et al., 2017) which learns the learning rate of parameters that enables to initialize and adapt any differentiable learner in a single step.

**Attacking a meta-learner.** To obtain robustness on few-shot tasks, Adversarial Querying (AQ) (Goldblum et al., 2020) proposes to generate attacks with only the query examples. The AQ employs the project gradient descent attack (PGD (Madry et al., 2018)), which is a class-wise attack that maximizes the cross-entropy on a given query image as follows,

$$\delta^{t+1} = \Pi_{B(x^q, x^q + \epsilon)}\left(\delta^t + \gamma \mathtt{sign}\left(\nabla_{\delta^t} \mathcal{L}_{\mathrm{CE}}\left(h_{\mu^\tau}(f_{\theta^\tau}(x^q + \delta^t)), y^q\right)\right)\right), \tag{3}$$

where $x^q$ and $y^q$ is a query image and its label of task $\tau$, respectively, $B(x^q, x^q + \epsilon)$ is the $l_\infty$ norm-ball around $x^q$ with radius $\epsilon$, $\gamma$ is step size of the attack, $\delta$ is perturbation and the cross entropy loss ($\mathcal{L}_{\mathrm{CE}}$) is calculated on the inner updated parameters ($\theta^\tau, \mu^\tau$).

**Robust training loss.** Various adversarial training methods have been proposed to enhance the model's robustness to adversarial attacks (Appendix A.1). Among them, we adapt the TRADES (Zhang et al., 2019) loss to improve robustness. TRADES proposes to regularize the model's outputs on the clean and adversarial examples with Kullback-Leibler divergence (KLD) as follows:

$$\mathcal{L}_{\mathrm{TRADES}} = \mathcal{L}_{\mathrm{CE}}\left(h_{\mu^\tau}(f_{\theta^\tau}(x^q)), y^q\right) + \beta \max_{\delta \in B(x^q, x^q + \epsilon)} \mathcal{L}_{\mathrm{KL}}\left(h_{\mu^\tau}(f_{\theta^\tau}(x^q)) || h_{\mu^\tau}(f_{\theta^\tau}(x^q + \delta))\right), \tag{4}$$

where $\mathcal{L}_{\mathrm{CE}}$ is cross-entropy loss on clean examples, $\mathcal{L}_{\mathrm{KL}}$ is KLD loss between clean and adversarial logit to obtain robustness, and $\beta$ is a regularizer to control the trade-off between clean accuracy and robustness which normally set as $6.0$. In our framework, we calculate the adversarial loss on query sets $(x^q, y^q)$, which are different instances used in inner adaptation, to meta-learn robust representations in the meta-optimization phase.

## 3.2 ADVERSARIAL META-LEARNING WITH SELF-SUPERVISED LEARNING

**Bi-level parameter augmentation in adversarial meta-learning.** In recent self-supervised learning (Chen et al., 2020; He et al., 2020; Grill et al., 2020), image augmentation is applied to produce multiple views of the same instance, which is used to learn non-linear transformation representation space that leads to learning good quality of the visual representations. Motivated by the self-supervised learning concept, to have transferable robustness in meta-learning, we propose a

---

**Algorithm 1** Transferable robust meta learning via bilevel attack (TROBA)

---

**Require:** Dataset $\mathbf{D}$, transformation function $t \sim \mathcal{T}$
**Require:** Encoder $f$, parameter of encoder $\theta$, classifier $h$, parameter of classifier $\mu$
**Require:** adversary $\mathcal{A}(\texttt{base}, \texttt{target}, \texttt{parameter})$
  **while** not done **do**
  Sample tasks $\{\tau\}$, Support set $\mathbf{S}(x^s, y^s)$, Query set $\mathbf{Q}(x^q, y^q)$
    **for** $i = 1, \cdots$ **do**
      Transform input $t_1(x^s), t_2(x^s)$
      Fine-tune model with $t_1(x^s), y^s$ and updates parameter $\theta_1^\tau$
      Fine-tune model with $t_2(x^s), y^s$ and updates parameter $\theta_2^\tau$
      Generate adversarial examples
      $t_1(x^q)^{adv} = \mathcal{A}(t_1(x^q), t_2(x^q), \theta_1^\tau), t_2(x^q)^{adv} = \mathcal{A}(t_2(x^q), t_1(x^q), \theta_2^\tau)$
      Calculate losses on query set images
      $\mathcal{L}_{\text{TRADES}_1} = \mathcal{L}_{\text{CE}}(h_\mu(f_{\theta_1^\tau}(t_1(x^q))), y^q) + \mathcal{L}_{\text{KL}}(f_{\theta_1^\tau}(t_1(x^q)), f_{\theta_1^\tau}(t_1(x^q)^{adv}))$
      $\mathcal{L}_{\text{TRADES}_2} = \mathcal{L}_{\text{CE}}(h_\mu(f_{\theta_2^\tau}(t_2(x^q))), y^q) + \mathcal{L}_{\text{KL}}(f_{\theta_2^\tau}(t_2(x^q)), f_{\theta_2^\tau}(t_2(x^q)^{adv}))$
      $\mathcal{L}_{\texttt{self-sup}} = \mathcal{L}_{\texttt{similarity}}(f_{\theta_1^\tau}(t_1(x^q)^{adv}), f_{\theta_2^\tau}(t_2(x^q)^{adv}))$
      $\mathcal{L}_{\texttt{Our}} = \mathcal{L}_{\text{TRADES}_1} + \mathcal{L}_{\text{TRADES}_2} + \mathcal{L}_{\texttt{self-sup}}$
      Compute gradient $g^\tau = \nabla_{\theta_1^\tau, \theta_2^\tau} \mathcal{L}_{\texttt{Our}}$
    **end for**
    Update model parameters
    $\theta, \mu \leftarrow \theta, \mu - \frac{\alpha}{\tau} \sum g^\tau$
  **end while**

---

bilevel parameter augmentation with self-supervised learning. Bilevel parameter augmentation enables the model to adapt the view-specific projected latent space to set of augmented samples of the given instance. Specifically, to generate augmented parameters of the encoder, we first generate multiple views of images with a stochastic data augmentation function $t$ that is randomly selected from the augmentation set $\mathcal{T}$, including random crop, random flip, random color distortion, and random grey scale as Zbontar et al. (2021). We then apply two random augmentations $t_1, t_2 \sim \mathcal{T}$ to images from both support set ($\mathbf{S} = \{t_1(x^s), t_2(x^s), y^s\}$) and query set ($\mathbf{Q} = \{t_1(x^q), t_2(x^q), y^q\}$).

Then, we generate multiple views of the shared parameters ($\theta_1^\tau$ and $\theta_2^\tau$) which are adapted parameters of encoder with differently transformed support sets ($\mathbf{S}_\tau = \{t_1(x^s), t_2(x^s), y^s\}$) as shown in Figure 1. Overall, we introduce parameter-level augmentation along with image-level augmentation to form a different view of single instances in the meta-learning framework, which we refer to as *bilevel parameter augmentation*.

**Bilevel attack with dynamic instance-wise attack.** On top of bilevel parameter augmentation, we propose a bilevel attack with a dynamic instance-wise attack to obtain generalized robustness in few-shot tasks. We redesign the instance-wise attack introduced in the self-supervised adversarial learning (Kim et al., 2020; Jiang et al., 2020), which generates adversaries by maximizing the instance classification loss in Equation 3. Specifically, we apply an instance-wise attack on our meta-learning framework, by generating adversaries that maximize the difference between the representations of the augmented samples of the same instance obtained by the encoder whose parameters are adapted to each view, as follows:

$$\delta_1^{t+1} = \Pi_{B(x^q, x^q + \epsilon)}\left(\delta_1^t + \alpha \texttt{sign}\left(\nabla_{\delta_1^t} \mathcal{L}_{\texttt{similarity}}(f_{\theta_1^\tau}(t_1(x^q) + \delta_1^t), f_{\theta_1^\tau}(t_2(x^q)))\right)\right),$$
$$\delta_2^{t+1} = \Pi_{B(x^q, x^q + \epsilon)}\left(\delta_2^t + \alpha \texttt{sign}\left(\nabla_{\delta_2^t} \mathcal{L}_{\texttt{similarity}}(f_{\theta_2^\tau}(t_2(x^q) + \delta_2^t), f_{\theta_2^\tau}(t_1(x^q)))\right)\right),$$
(5)

where $\delta_1, \delta_2$ are generated perturbations to maximize the difference between features from each bilevel augmented encoder $f(\theta_1^\tau)$ and $f(\theta_2^\tau)$ respectively. The maximized loss $\mathcal{L}_{\texttt{similarity}}$ is the instance-wise classification loss used in adversarial self-supervised learning (Kim et al., 2020). We use the differently transformed query counterpart sets as a target for dynamic instance-wise attack and calculate perturbations with the parameter of the augmented encoder.

**Adversarial meta-learning with bilevel attack.** We now present a framework to learn transferable robust representations via bilevel attack for unseen domains. The gradient ($g$) is calculated to minimize our proposed objective as follows:

$$g = \nabla_{\theta_1^\tau, \theta_2^\tau, \mu} \mathcal{L}_{\texttt{Our}}(h_\mu, f_{\theta_1^\tau}, f_{\theta_2^\tau}, t_1(x^q), t_1(x^q)^{adv}, t_2(x^q), t_2(x^q)^{adv}, y^q),$$
(6)

Table 1: Results of transferable robustness in 5-way 5-shot unseen domain tasks that are trained on 5-way 5-shot CIFAR-FS. Rob. stands for accuracy(%) that is calculated with PGD-20 attack ($\epsilon = 8./255.$, step size=$\epsilon/10$). Clean stands for test accuracy(%) of clean images. All models are trained with PGD-7 attacks on ResNet12.

| | CIFAR-FS → | | | | | | | | | |
| | Mini-ImageNet | | Tiered-ImageNet | | CUB | | Flowers | | Cars | |
| | Clean | Rob. | Clean | Rob. | Clean | Rob. | Clean | Rob. | Clean | Rob. |
|---|---|---|---|---|---|---|---|---|---|---|
| MAML (Finn et al., 2017) | 44.85 | 6.21 | **61.19** | 2.48 | 48.41 | 3.46 | **67.76** | 5.73 | **43.94** | 5.31 |
| ADML (Yin et al., 2018) | 28.66 | 6.53 | 40.06 | 11.36 | 31.18 | 5.21 | 39.36 | 11.26 | 27.43 | 3.18 |
| AQ (Goldblum et al., 2020) | 33.09 | 3.32 | 37.41 | 5.05 | 38.37 | 4.10 | 60.14 | 11.03 | 36.83 | 4.47 |
| RMAML (Wang et al., 2021) | 28.05 | 6.65 | 29.54 | 9.30 | 30.24 | 5.67 | 42.91 | 10.79 | 31.72 | 5.56 |
| Ours | **45.82** | **24.12** | 51.46 | 30.06 | **48.56** | **25.23** | 66.49 | **42.16** | 38.29 | **19.43** |

where $\mathcal{L}_{\mathtt{Our}}$ is the meta-objective loss to obtain generalized robustness, $h_\mu$ is a meta-initialized classifier, and $f_{\theta_1}$ and $f_{\theta_2}$ are bilevel augmented encoder for each view. Further, the $\mathcal{L}_{\mathtt{Our}}$ consists of adversarial loss, i.e., TRADES (Zhang et al., 2019) loss, and self-supervised loss as follow,

$$\mathcal{L}_{\mathtt{Our}} = \sum_{n=1}^{2} \left[ \mathcal{L}_{\mathtt{CE}}(l_n, y^q) + \mathcal{L}_{\mathtt{KL}}(l_n^{adv}, l_n) \right] + \mathcal{L}_{\mathtt{self-sup}}(z_1^{adv}, z_2^{adv}), \tag{7}$$

where $z_n = f_{\theta_n^\tau}(t_n(x^q))$ and $l_n = h_\mu(z_n)$ are a feature and a logit of each multi-view instance with augmented encoder $f_{\theta_n^\tau}$ and meta-initialized classifier $h_\mu$ respectively, $\mathcal{L}_{\mathtt{CE}}$ is a cross-entropy loss, $l_n^{adv}$ is a logit of an attacked image generated from our bilevel attack which is a dynamic instance-wise attack, $\mathcal{L}_{\mathtt{KL}}$ is a KL-divergence loss, and $\mathcal{L}_{\mathtt{self-sup}}$ is a cosine similarity loss between two differently augmented features. This sum of the cross-entropy and KL-divergence loss is the TRADES loss to learn robustness for each augmented encoder for each task (Equation 4). The crucial component here is the self-supervised loss which regularizes our model to have robust consistency between the features from the two different views, which helps it learn robust representations across any instances or augmentations, allowing it to achieve transferable robustness. The overall algorithm of TROBA is described in Algorithm 1.

## 4 EXPERIMENT

In this section, we first validate the robustness of our model in unseen domain few-shot learning tasks (Section 4.1). Then, we analyze our model through ablation experiments on the type of loss, attack, and augmentations (Section 4.2). Specifically, our models also show comparable transfer robustness to that of the self-supervised framework trained in a standard learning setting with large datasets, although we use a significantly smaller amount of data for pre-training (Section 4.3).

**Experimental Setup.** For meta-learning setting, we train our approach on ResNet12 with 5-way 5-shot images in both CIFAR-FS and Mini-ImageNet. We train our model with the BOIL (Oh et al., 2020) framework, and take a single step in both inner and outer optimization as done in (Li et al., 2017) during meta-training and the inner optimization of meta-testing steps. We adversarially train our model with $\ell_\infty$ PGD attacks with the epsilon of $8/255$, alpha of $2/255$ in 7 steps. We evaluate the robustness against $\ell_\infty$ PGD attacks with the epsilon of $8/255$ and 20 iterations for evaluation, following the standard procedure. The code will be available in Anonymous. More experimental details are in Appendix B.

### 4.1 RESULTS OF ADVERSARIAL ROBUSTNESS IN FEW-SHOT TASK

**Robustness in unseen domain.** Since our main goal is to achieve transferable robustness in unseen domains, we mainly validate our methods on unseen domain few-shot tasks. We meta-train our model on CIFAR-FS and meta-test on the benchmark datasets with different domains such as Mini-ImageNet, Tiered-ImageNet, CUB, Flowers, and Cars. As shown in Table 1, previous adversarial meta-learning methods have difficulty in achieving robustness on unseen domains. However, TROBA is able to show impressive transferable robustness in this cross-domain

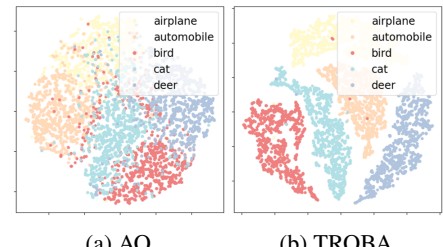

(a) AQ      (b) TROBA

Figure 2: Visualization of feature on unseen domains, CIFAR-10, where models are trained on CIFAR-FS.

Table 2: Results of transferable robustness in 5-way 5-shot unseen domain tasks that are trained on 5-way 5-shot Mini-ImageNet. Rob. stands for accuracy(%) that is calculated with PGD-20 attack ($\epsilon = 8./255.$). Clean stands for test accuracy(%) of clean images. All models are trained with PGD-7 attacks on ResNet12.

| | Mini-ImageNet → | | | | | | | | | |
| | CIFAR-FS | | Tiered-ImageNet | | CUB | | Flowers | | Cars | |
| | Clean | Rob. | Clean | Rob. | Clean | Rob. | Clean | Rob. | Clean | Rob. |
|---|---|---|---|---|---|---|---|---|---|---|
| MAML (Finn et al., 2017) | **66.75** | 12.97 | **65.33** | 13.10 | 52.82 | 4.46 | **71.01** | 4.86 | **43.66** | 2.77 |
| ADML (Yin et al., 2018) | 41.14 | 13.36 | 41.05 | 13.26 | 32.82 | 4.59 | 43.07 | 9.65 | 24.85 | 5.48 |
| AQ (Goldblum et al., 2020) | 61.97 | 30.73 | 47.61 | 14.21 | 45.64 | 13.19 | 65.40 | 25.01 | 37.29 | 8.85 |
| RMAML (Wang et al., 2021) | 37.94 | 10.59 | 30.49 | 8.24 | 27.30 | 6.26 | 42.52 | 13.08 | 37.76 | 5.43 |
| Ours | 65.45 | **36.51** | 59.64 | **29.73** | 53.70 | **20.64** | 69.84 | **36.49** | 42.25 | **14.42** |

Table 3: Comparison in the 5-shots seen domain tasks. All models are trained on CIFAR-FS and Mini-ImageNet, respectively, with PGD-7 attack in ResNet12. * stands for reported results in Wang et al. (2021).

| | CIFAR-FS | | Mini-ImageNet | |
| | Clean | Rob. | Clean | Rob. |
|---|---|---|---|---|
| ADML (Finn et al., 2017) | 53.06 | 22.45 | 26.72 | 6.81 |
| AQ (Goldblum et al., 2020) | 73.49 | 28.49 | 39.47 | 13.52 |
| RMAML (Wang et al., 2021)* | 57.95 | 35.30 | 43.98 | 21.47 |
| Ours (Li et al., 2017) | 64.90 | 43.34 | 47.56 | 18.18 |

Table 4: Results of TROBA with a different meta-learning frameworks in 5-shot tasks. All models are trained on CIFAR-FS and Mini-ImageNet, respectively, with PGD-7 attacks in ResNet12.

| | CIFAR-FS | | Mini-ImageNet | |
| | Clean | Rob. | Clean | Rob. |
|---|---|---|---|---|
| TROBA | | | | |
| +MAML (Finn et al., 2017) | 52.79 | 32.50 | 37.58 | 14.23 |
| +FOMAML (Finn et al., 2017) | 53.42 | 35.95 | 33.87 | 15.60 |
| +Meta-SGD (Li et al., 2017) | 64.90 | 43.34 | 47.56 | 18.18 |

task. It also obtains significantly better clean accuracy over the adversarial meta-learning baselines, while obtaining competitive clean accuracy to MAML. In particular, TROBA shows better robustness compared to baselines even though the distribution of the unseen domain is highly different from the distributions of the meta-trained dataset (i.e., CUB, Flowers, Cars).

Further visualization of the representations of the instance-wise adversarial samples from the unseen domains shows that our model is able to obtain well-separated feature space for attacked samples on this novel domain (CIFAR-10) even before adapting to it, while the previous adversarial meta-learning framework learns a feature space with large overlaps across the adversarial instances belonging to different classes (Figure 2). This suggests that the superior performance of our model (Table 1, 2) mainly comes from its ability to learn such transferable robust representations. In particular, TROBA has smoother loss surface to adversarial examples compared to the baseline, which is why TROBA could demonstrate better robustness in unseen domain (Figure 3).

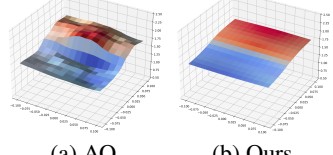

(a) AQ      (b) Ours

Figure 3: Loss surface of unseen domain (Mini-ImageNet)

**Robustness in seen domain.** Even though TROBA is designed to have transferable robustness in the unseen domain, our methods also show better robustness in seen domain few shot tasks compare to baselines, even with better clean accuracy (Table 3). In addition, TROBA shows smoother loss surface to adversarial examples which is also directly associated with better robustness and generalization (Figure 4). Our method is agnostic to the meta-learning approach, as shown in Table 4, which suggests that the type of meta-learning strategy is not the main factor in achieving the transferable robustness. We only update the encoder in the inner optimization for all meta-learning algorithms.

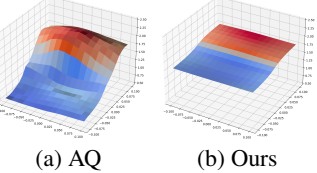

(a) AQ      (b) Ours

Figure 4: Loss surface of seen domain (CIFAR-FS)

## 4.2 ABLATION STUDY

To examine each component of our proposed methods, we conduct the ablation study on augmentation, loss, and attack. Through our ablation study, we verify the effectiveness of each component by their robustness on unseen domains.

**Bilevel parameter augmentation contributes to learn generalized features.** As shown in Table 5, image-only augmentation alone meaningfully contributes to learning generalized features

Table 5: Ablation study of our proposed bilevel augmentation. Test accuracy(%) on seen domain (CIFAR-FS) and unseen domains (Mini-ImageNet, Flower, Cars) of 5-way 5-shot task. Robustness is calculated with PGD-20 attack($\epsilon = 8./255.$, step size=$\epsilon/10$), clean stands for accuracy of clean images. All models are adversarially meta-trained on CIFAR-FS with attack step 3 due to computation costs.

| Augmentation level | | CIFAR-FS | | Mini-ImageNet | | Tiered-ImageNet | | Cars | |
|---|---|---|---|---|---|---|---|---|---|
| Image Aug. | Parameter Aug. | Clean | PGD $\ell_\infty$ | Clean | PGD $\ell_\infty$ | Clean | PGD $\ell_\infty$ | Clean | PGD $\ell_\infty$ |
| ✔ | - | 63.10 | 36.98 | 39.54 | 15.08 | 51.57 | 25.05 | 38.99 | 14.36 |
| ✔ | ✔ | **65.82** | **41.39** | **44.64** | **15.75** | **53.25** | **28.05** | **40.08** | **16.88** |

(a) Clean Accuracy

(b) Robustness

Figure 6: Ablation on meta-objectives in TROBA. Test accuracy(%) on the seen domain (CIFAR-FS) and unseen domains (Mini: Mini-ImageNet, Tiered: Tiered-ImageNet, Flower, CUB, Cars) of 5-way 5-shot task. Legends denote the meta-objectives loss that is used to train the model. All models are adversarially meta-trained on CIFAR-FS with attack step 3 due to computation overhead. (a) Clean accuracy stands for the accuracy of clean images. (b) Robustness is calculated with PGD-20 attack($\epsilon = 8./255.$, step size=$\epsilon/10$).

for unseen domains. However, when we apply parameter augmentation on top of the image augmentation, the model achieves significantly better clean and robust accuracy than the model trained with image-only augmentation, especially in the seen domain. This suggests that the bilevel parameter augmentation is effective in learning consistent representations across tasks and views.

To support our claim, we calculate the Centered Kernel Alignment (CKA) (Kornblith et al., 2019) value, which measures the similarity between representations (When representations are identical, the CKA is 1). As shown in Figure 5, when bilevel parameter augmentation is applied, features from the augmented parameters are more dissimilar than features with image augmentation only. These results show that our bilevel parameter augmentation may generate more different multi-views of the same instances which helps learn invariant representations across views, that help it to achieve generalizable robustness.

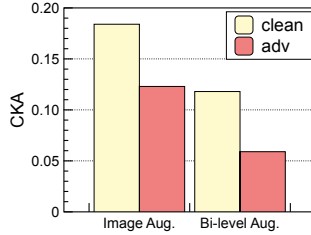

Figure 5: Effect of augmentation

**Self-supervised loss regularized to learn generalized features.** TROBA leverages both adversarial loss and self-supervised loss in meta-objective; specifically, it uses TRADES loss (Equation 4) and cosine similarity loss between representations of differently bilevel augmented views, as shown in Equation 7. The adversarial loss is calculated independently in each bilevel augmented network to enhance the robustness on each training sample. On the other hand, the self-supervised loss is computed between the representations of each bilevel augmented encoder to enforce the consistency across features for samples attacked with our bilevel attack, which helps it to obtain a consistent representation space across perturbations and instances, which helps with its generalization (Figure 6). Notably, the self-supervised loss has a larger contribution when we conduct transfer learning to unseen domains with larger data (Appendix D.2).

**Bilevel attack makes the model to be robust on unseen domain attacks.** We further analyze the effect of our bilevel instance-wise attack compared to class-wise attack in Table 6. We observe that adversarial examples that are attacked with instance-wise attack make the model more robust in unseen domains compared to class-wise attack. Specifically, instance-wise attack generates adversaries that have larger difference to clean examples in the representation level, and thus can be thought as a stronger attack. To demonstrate the effectiveness of instance-wise attack, we calculate CKA (Kornblith et al., 2019) between clean and adversarial features from each bilevel augmented parameters. As

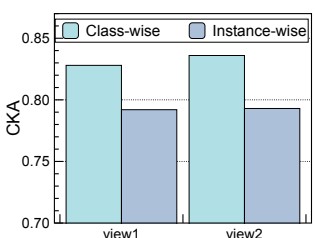

Figure 7: Effect of type of attack

Table 6: Ablation study of our proposed bilevel attack. Test accuracy(%) on seen domain (CIFAR-FS) and unseen domains (Mini-ImageNet, Tiered-ImageNet, Flower, Cars) of 5-way 5-shot task. Clean stands for accuracy of clean images. Rob. stands for robust accuracy that is calculated with PGD-20 attack($\epsilon = 8./255.$). All models are adversarially meta-trained on CIFAR-FS with attack step 3 due to computation costs.

| Attack type | CIFAR-FS | | Mini-ImageNet | | Tiered-ImageNet | | Flowers | | CUB | |
|---|---|---|---|---|---|---|---|---|---|---|
| | Clean | Rob. | Clean | Rob. | Clean | Rob. | Clean | Rob. | Clean | Rob. |
| Bi-level class-wise | **66.69** | 40.48 | 42.15 | **17.13** | **53.91** | 27.41 | 69.66 | 38.83 | 50.01 | 21.2 |
| Bi-level instance-wise | 65.82 | **41.39** | **44.64** | 15.75 | 53.25 | **28.05** | **70.08** | **41.52** | **50.78** | **22.44** |

Table 7: Experiments results in robust full-finetuning of TROBA and the state-of-the-art adversarial self-supervised learning (SSL) models. While TROBA is trained on CIFAR-FS, other models are trained on the CIFAR-100. TROBA is pre-trained with bilevel attacks with 3 steps due to computational overhead, others are pre-traiend with PGD-7 attacks. All models are trained on ResNet18. We evaluate all models with PGD-20 steps and AutoAttack (AA) (Croce & Hein, 2020) with $\epsilon$=8/255.

| | Method | CIFAR-10 | | | STL-10 | | | CIFAR-100 | | |
|---|---|---|---|---|---|---|---|---|---|---|
| | | Clean | PGD-20 | AA | Clean | PGD-20 | AA | Clean | PGD-20 | AA |
| SSL | RoCL (Kim et al., 2020) | 76.76 | 50.72 | 45.52 | 60.44 | 31.90 | 27.38 | 51.91 | 27.77 | 22.79 |
| | ACL (Jiang et al., 2020) | 75.99 | 50.35 | 45.50 | 63.46 | 30.24 | 25.73 | 51.91 | 27.77 | 22.79 |
| | BYORL (Gowal et al., 2020) | 76.39 | 50.51 | 45.37 | 62.85 | 28.15 | 24.23 | 52.37 | 28.09 | 23.11 |
| | **Ours (3 steps)** | 74.26 | 49.38 | 44.31 | 53.46 | 32.65 | 28.96 | 50.23 | 27.05 | 21.96 |

shown in Figure 7, instance-wise attack produces more difficult adversarial examples that are highly dissimilar from clean instances. However, when the parameter is augmented with bilevel parameter augmentation, the class-wise attack also can show transferable robustness since self-supervised loss supports it to obtain generalized robustness.

## 4.3 TRANSFERABLE ROBUSTNESS IN DIFFERENT DOMAINS

To demonstrate the power of our adversarially transferable meta-trained model, we further evaluate our model on a standard transfer learning scenario that employs full data to fully train the encoder with the linear layer on top of it. Specifically, we want to evaluate the generalizable robustness of the representations learned by our encoder against a self-supervised learning model trained with a large amount of data. We evaluate our model on the seen domain, CIFAR-100, as well as on two unseen domains, which are CIFAR-10 and STL-10 respectively. As shown in Table 7, our model shows comparable clean accuracy and robustness in the unseen domains despite the difference in the amount of data used to train the model. Our model is pre-trained with scarce data, and we have even reduced the number of the steps for the bilevel attack to 3 steps to reduce the computational cost, but obtains competitive performance to the model trained with larger data. The experimental results suggest that we may use our method as a means of pretraining the representations to ensure robustness for a variety of applications, when the training data is scarce.

## 5 CONCLUSION

We proposed a novel adversarial self-supervised meta-learning framework that can learn transferable robust representations using only a few data via bilevel attack, which introduces a novel bilevel parameter augmentation along with dynamic instance-wise attack. Specifically, the bilevel attack leverages self-supervised learning to effectively generate robust representation of multi-views with differently augmented encoder, which allows learning non-linear transformation task-adaption that brings good robust generalization power. While previous adversarial meta-learning methods are extremely vulnerable to unseen domains, our model learned generalized robust representations which can demonstrate impressive transferable robustness on few-shot tasks in unseen domains. Moreover, we validate our models on larger data in unseen domains which shows comparable robust representations with self-supervised learning (SSL) model with much fewer data. We hope that our work inspires adversarial meta-learning to obtain a good robust representations only using a few data.

## REPRODUCIBILITY

- **Datasets.** We use CIFAR-FS, Mini-ImageNet, Tiered-ImageNet, CUB, Flower, and Cars for few-shot learning tasks. Further, we use CIFAR-10, CIFAR-100, STL-10 for standard image classification tasks in transfer learning. More details are in Supplementary B.1.
- **Meta-train.** We trained our models on CIFAR-FS and Mini-ImageNet with ResNet12 and ResNet18 as the base encoder. All models are trained on 5-way 5-shot support set images and 5-way 15-shot query images. More details are in Supplementary B.2.
- **Meta-test.** We evaluate our models on few-shot learning tasks as described in Supplementary B.4. For adversarial setting for few-shot robustness is written in Supplementary B.5.
- **Table 1, 2, 3.** Please see Section 4.1 and Supplementary B.2, B.4 to reproduce the results in those Table.
- **Table 4, 9.** Please see Section 4.1 to reproduce the results, and more details for hyperparameters are in Supplementary B.3. For adversarial setting, see Supplementary B.5.
- **Table 5.** Please see Section 4.2 and Supplementary C.1 to reproduce the Table 5. Detailed algorithm for bilevel parameter augmentation and image-only augmentations we utilized are described in Algorithm 1 and Algorithm 2, respectivley.
- **Table 6.** Please see Supplementary C.2 to reproduce the Table 6. Bilevel attack is explained in Section 3.2, 4.2.
- **Table 7, 10, 12.** Please see Supplementary C.5 to check the baselines, and see Supplementary B.5 to reproduce the results.
- **Table 8.** Please see Supplementary C.4 to reproduce the results.
- **Table 11.** Please see Supplementary D.2 to reproduce the results.
- **Table 13.** Please see Supplementary E.2 to reproduce the results.
- **Table 14.** Please see Supplementary F to reproduce the results.
- **Figure 4, 3.** Please see Supplementary G to reproduce the results.
- **Figure 5, 7.** Please see Supplementary C.3 to reproduce the results.

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

# Supplementary Material

## Few-Shot Transferable Robust Representation Learning via Bilevel Attacks

## A  RELATED WORKS

### A.1  ADVERSARIAL LEARNING

The vulnerability of deep neural network (DNN) to imperceptible small perturbation on the input is a well-known problem as observed in previous works (Biggio et al., 2013; Hendrycks & Dietterich, 2019; Szegedy et al., 2013). To overcome the adversarial vulnerability, many attack-based approaches for constructing perturbed examples (Goodfellow et al., 2015; Carlini & Wagner, 2017; Papernot et al., 2016) have appeared. On the other hand, Madry et al. (2018) proposes a defense-based approach against adversarial examples. Madry et al. (2018) utilizes a project gradient descent (PGD) in the perspective of robust optimization, which maximizes the loss in the inner-maximization loops while minimizing the overall loss on tasks in outer-minimization loops, the so-called min-max formulation. Zhang et al. (2019) theoretically shows the trade-off between clean accuracy and robustness in adversarial training. To improve both clean and robust accuracy, TRADES (Zhang et al., 2019) introduces regularized surrogate loss. Especially, the Kullback-Leibler divergence (KLD) in TRADES (Zhang et al., 2019) helps to enhance the robustness by enforcing consistency between representations of clean and adversarial examples. Afterward, significant advances in adversarial robustness have emerged. Kim et al. (2020); Jiang et al. (2020) proposes a self-supervised adversarial learning mechanism coined with contrastive learning to obtain a robust representation without explicit labels. Since a larger dataset is essential to have better adversarial robustness, Shafahi et al. (2019) leverages transfer learning to transfer learned robust representations into new target domains with only a few data. Goldblum et al. (2020) proposes robust supervised meta-learners with adversarial query images in few-shot classification tasks. However, previous works still have difficulty in obtaining generalized robustness on multiple datasets.

### A.2  SELF SUPERVISED LEARNING

Conventional adversarial learning mechanisms in a supervised manner require the label information which needs expensive human labeling annotations. Self-supervised learning makes the neural networks possible to learn comparable representations to supervised representations, even it does not leverage labels (Grill et al., 2020). Many previous works focus on learning consistent representations to different distortions in the input (Koch et al., 2015; Chen et al., 2020; He et al., 2020; Tian et al., 2020). To learn the distortion-invariant representations, they enforce consistency between representations of two differently augmented inputs with the same instance-level identity. Especially, Chen et al. (2020) employs contrastive learning to maximize agreement only between positive pairs in mini-batch while negative pairs are handled as the opposite. In advance, other works introduce asymmetry into network architecture or parameter update (Chen & He, 2021; Grill et al., 2020) to improve performance. However, the existence of trivial solutions derived from asymmetry leaves room to improve. Zbontar et al. (2021) achieves comparable performance by introducing redundancy reduction terms in the training objectives, even it does not require additional asymmetric networks or large batches.

### A.3  SELF-SUPERVISED ADVERSARIAL LEARNING

Utilizing the advantages of self-supervised learning, adversarial training mechanisms in a self-supervised manner have emerged to learn robust representations without relying on label information. Recent works leverage contrastive learning to obtain robust representation in a self-supervised manner (Kim et al., 2020; Jiang et al., 2020). Kim et al. (2020) first devises the instance-wise adversarial perturbation, which does not require explicit labels during the attack, and utilizes those perturbed examples in maximizing contrastive loss. Jiang et al. (2020) introduces a dual stream with optimizing two contrastive losses against four augmented views, which are computed between clean views and adversarial images, respectively. However, these approaches highly rely on large

batch sizes to effectively create positive and negative samples for the contrastive learning framework. Gowal et al. (2020) injects adversarial examples on top of the BYOL framework (Grill et al., 2020) to achieve robustness to avoid the restrictions on large batch sizes. Although existing restrictions on large batch sizes or image augmentation have been relieved during extensive development in self-supervised adversarial training, obtaining robustness with scarce data is still difficult, even in a supervised manner.

# B  EXPERIMENTAL DETAILS

## B.1  DATASET

For meta-training, we use CIFAR-FS (Bertinetto et al., 2019) and Mini-ImageNet (Russakovsky et al., 2015). CIFAR-FS and Mini-ImageNet consist of 100 classes which are 64, 16, and 20 for meta-training, meta-validation, and meta-testing, respectively. We validate our model on 6 benchmark few-shot datasets: CIFAR-FS (Bertinetto et al., 2019), Mini-ImageNet (Russakovsky et al., 2015), Tiered-ImageNet (Russakovsky et al., 2015), Cars, CUB and VGG-Flower, for few-shot classification and 3 additional benchmark standard image classification datasets: CIFAR-10, CIFAR-100, and STL-10, for robust transferability. CIFAR-10 and CIFAR-100 consist of 50,000 training images and 10,000 test images with 10 and 100 classes, respectively. All images are used with $32{\times}32{\times}3$ resolution (width, height, and channel) for meta-training. Especially, we apply *TorchMeta*[1] library to load the few-shot datasets into our frameworks.

## B.2  META-TRAIN

We meta-train ResNet12 and ResNet18 as the base encoder network on CIRAR-FS and Mini-ImageNet. All models are meta-trained with tasks consist of 5-way 5-shot support set images and 5-way 15shot query set images, and meta-validated with only clean tasks consist of 5-way 1-shot support set images and 5-way 15-shot query set images. Especially, we train the model with randomly selected 200 tasks and validate the model with randomly selected 100 tasks. For optimization, we meta-train our models with 300 epochs under SGD optimizer with weight decay 1e-4. For data augmentation, we use random crop with 0.08 to 1.0 size, color jitter with probability 0.8, horizontal flip with probability 0.5, grayscale with 0.2, gaussian blur with 0.0, and solarization probability with 0.0 to 0.2. We exclude normalization for adversarial training.

In the case of adversarial learning, we use our proposed bilevel attack with 3 steps and 7 steps. To generate adversaries with query set images, we take the gradient step within $l_\infty$ norm ball with $\epsilon$ = 8.0/255.0 and $\alpha$ = 2.0/255.0 to maximize the similarity with target instance. To obtain robust representation, we utilize an adversarial loss and self-supervised loss which are TRADES (Zhang et al., 2019) with a regularization hyperparameter of 6.0 and cosine similarity loss, respectively. Overall TROBA model figure is shown in Figure 8.

Three different meta-learning frameworks are leveraged to train our model, which are MAML (Finn et al., 2017), FOMAML (Finn et al., 2017) and Meta-SGD (Li et al., 2017). Specifically, we only update the encoder parameters in inner optimization for all three meta-learning strategies. Detailed hyperparameters for meta-train and meta-test will be described in B.3.

## B.3  HYPERPARAMETER DETAILS OF EACH META-LEARNING FRAMEWORKS

**MAML**   We take a single step for both inner optimization and outer optimization to meta-train ResNet12 on CIFAR-FS and Mini-ImageNet. We use the same learning rate for both datasets, which are 0.3 and 0.08 for outer optimization and inner optimization, respectively. For both dataset, we use batch size 4.

**FOMAML**   To reduce the computational cost, we try to adapt FOMAML (Finn et al., 2017), which is the first-order approximation of MAML (Finn et al., 2017). For ResNet18, we use a single step in both inner optimization and outer optimization, and use the learning rates 0.3 and 0.4 in outer optimization and inner optimization, respectively. For ResNet12, we use 3 steps for inner

---

[1]https://github.com/tristandeleu/pytorch-meta

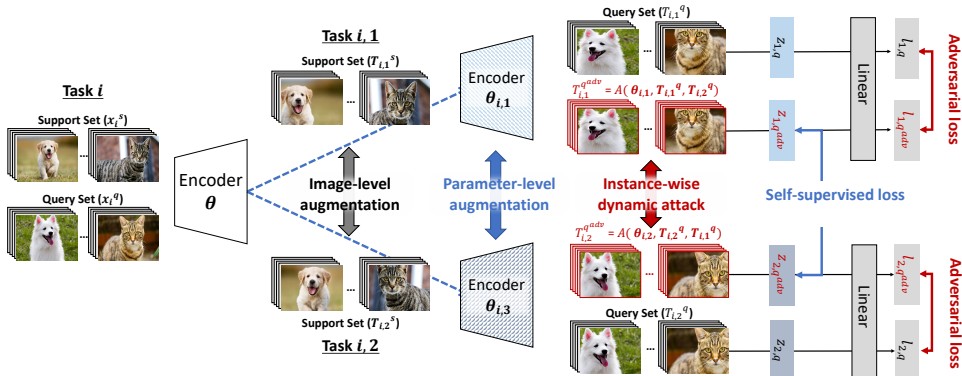

Figure 8: **Overall model figure of TROBA.**

optimization, and 1 step for outer optimization. We use learning rate 0.3 and 0.2 for outer optimization and inner optimization, respectively. For both dataset, we use batch size 4.

**META-SGD** To learn quickly, we use the Meta-SGD (Li et al., 2017) with the single step. We use a single step in inner optimization and use the $0.005$ inner learning rate. For outer loop, we use $0.005$ outer learning rate for CIFAR-FS. For Mini-ImageNet, we use a same step size as CIFAR-FS but with different inner learning rate, $0.001$, and outer optimization learning rate $0.001$. For both dataset, we use batch size 4.

## B.4 META-TEST

The trained models are evaluated with 400 randomly selected tasks from test set, where each task consists of 5-way 5-shot support set images and 5-way 15-shot query set images. We use a single step in both inner optimization and outer optimization. We especially use same learning rate and meta step size as the model is meta-trained.

## B.5 ADVERSARIAL EVALUATION

**FEW-SHOT ROBUSTNESS** We validate the robustness of our trained models against two types of attack, which are PGD (Madry et al., 2018) and AutoAttack (Croce & Hein, 2020). All $l_\infty$ PGD attacks are conducted with the norm ball size $\epsilon = 8./255.$, step size $\alpha = 8./2550.$, and with 20 steps of inner maximization. AutoAttack[2] is a combination of 4 different types of attacks (i.e., APGD-CE, APGD-T, FAB-T, and Square). We use the standard version of AutoAttack in the test time.

**SELF-SUPERVISED ROBUST LINEAR EVALUATION** To compare TROBA with self-supervised pre-trained models, we apply robust full-finetuning. In robust full-finetuning, the parameters of the entire network, including the feature extractor and the fc layer, are trained with adversarial examples. We generate perturbed examples with $l_\infty$ PGD-10 attack with $\epsilon = 8./255.$ and step size $\alpha = 2./255.$ in training. All adversarially full-finetuned models are evaluated against $l_\infty$ PGD-20 attack ($\epsilon = 8./255., \alpha = 8./2550.$) and AutoAttack (Croce & Hein, 2020). Especially, in comparisons with self-supervised models, we pre-train ResNet18 based on FOMAML (Finn et al., 2017), which is the first-order approximation of MAML (Finn et al., 2017), and apply bilevel attacks with 3 steps to reduce the computational cost. Other self-supervised models are pre-trained with PGD-7 attacks. For optimization, we fine-tune the pre-trained models for 110 epochs with batch size 128 under SGD optimizer with weight decay 5e-4, where Pang et al. (2022) demonstrated as optimal for robust full-finetuning on CIFAR datasets.

---

**Algorithm 2** Transferable robust meta learning via image-only augmentation

---

**Require:** Dataset $\mathbf{D}$, transformation function $t \sim \mathcal{T}$
**Require:** Encoder $f$, parameter of encoder $\theta$, classifier $h$, parameter of classifier $\mu$
**Require:** Adversary $\mathcal{A}(\texttt{base}, \texttt{target}, \texttt{parameter})$
  **while** not done **do**
  Sample tasks $\{\tau\}$, Support set $\mathbf{S}(x^s, y^s)$, Query set $\mathbf{Q}(x^q, y^q)$
    **for** $\tau = 1, \cdots,$ **do**
      Transform input $t_1(x^s), t_2(x^s)$
      Fine-tune model with $t_1(x^s), t_2(x^s), y^s$ and updates parameter $\theta^\tau$
      Generate adversarial examples
      $t_1(x^q)^{adv} = \mathcal{A}(t_1(x^q), t_2(x^q), \theta^\tau), t_2(x^q)^{adv} = \mathcal{A}(t_2(x^q), t_1(x^q), \theta^\tau)$
      Calculate losses on query set images
      $\mathcal{L}_{\text{TRADES}_1} = \mathcal{L}_{\text{CE}}(h_\mu(f_{\theta^\tau}(t_1(x^q))), y^q) + \mathcal{L}_{\text{KL}}(f_{\theta^\tau}(t_1(x^q)), f_{\theta^\tau}(t_1(x^q)^{adv}))$
      $\mathcal{L}_{\text{TRADES}_2} = \mathcal{L}_{\text{CE}}(h_\mu(f_{\theta^\tau}(t_2(x^q))), y^q) + \mathcal{L}_{\text{KL}}(f_{\theta^\tau}(t_2(x^q)), f_{\theta^\tau}(t_2(x^q)^{adv}))$
      $\mathcal{L}_{\texttt{self-sup}} = \mathcal{L}_{\texttt{similarity}}(f_{\theta^\tau}(t_1(x^q)^{adv}), f_{\theta^\tau}(t_2(x^q)^{adv}))$
      $\mathcal{L}_{\texttt{Our}} = \mathcal{L}_{\text{TRADES}_1} + \mathcal{L}_{\text{TRADES}_2} + \mathcal{L}_{\texttt{self-sup}}$
      Compute gradient $g^\tau = \nabla_{\theta^\tau} \mathcal{L}_{\texttt{Our}}$
    **end for**
    Update model parameters
    $\theta, \mu \leftarrow \theta, \mu - \frac{\alpha}{\tau} \sum g^\tau$
  **end while**

---

## C   EXPERIMENTAL DETAILS OF ABLATION STUDIES

### C.1   ABLATION STUDY OF BILEVEL PARAMETER AUGMENTATION

To demonstrate how bilevel parameter augmentation is more effective than image augmentation in adversarial self-supervised meta-learning, we experiment in the same environment except for parameter augmentation in inner adaptation. Specifically, we generate augmented parameters of the encoder adapted with two differently transformed support set images simultaneously, while TROBA augments parameters independently for each augmented view. A detailed algorithm for applying image-only augmentation in adversarial self-supervised meta-learning is described in Algorithm 2. Experiment results are reported in Section 4.2.

### C.2   ABLATION STUDY OF BILEVEL ATTACK

The bilevel attack is based on the instance-wise attack (Kim et al., 2020) which does not require label information to generate adversaries, while the class-wise attack utilizes label to maximize the cross-entropy loss in the inner maximization of Equation 3. The bilevel class-wise attack is applied with the bilevel augmented parameters as done in the bilevel attack. We use $l_\infty$ PGD attack with strength $8./255.$, step size $2./255.$, and the same number of iterations with bilevel attacks in all comparisons in the main paper.

### C.3   CKA ANALYSIS

We calculate CKA to demonstrate that bilevel attack constructs more confusing perturbed examples than class-wise attack, which helps in obtaining robust transferability. Specifically, we randomly selected 600 samples for the same tasks in charge of 10% for the entire dataset of CIFAR-FS that has 6,000 images per class. With randomly sampled tasks and the same pre-trained models, the parameter of the encoder is adapted with two differently transformed support set images as done in meta-training (i.e., bilevel parameter augmentation). Here, we generate adversaries with class-wise attack and bilevel attack independently with adapted encoders for each multiple view. Then we calculate CKA between the features of the clean and adversarial query set images with the same view, and average the CKA values over selected tasks. Results are reported in Section 4.2.

---

[2]https://github.com/fra31/auto-attack

Table 8: Ablation study on adversarial loss in meta-objectives of TROBA. Test accuracy(%) on benchmark data sets for 5-shots. Robustness is calculated with PGD-20 attack ($\epsilon = 8./255.$, step size=$\epsilon/10$), clean is for clean images. All models are adversarially meta-trained on CIFAR-FS, with ResNet18 as the base encoder.

| | Mini-ImageNet | | Tiered-ImageNet | | CUB | | Flower | | Cars | |
|---|---|---|---|---|---|---|---|---|---|---|
| Adversarial Loss | Clean | PGD $\ell_\infty$ | Clean | PGD $\ell_\infty$ | Clean | PGD $\ell_\infty$ | Clean | PGD $\ell_\infty$ | Clean | PGD $\ell_\infty$ |
| AT (Madry et al., 2018) | **33.78** | 7.99 | 38.35 | 12.95 | 37.00 | 10.08 | 42.28 | 22.19 | 30.93 | 9.59 |
| TRADES (Madry et al., 2018) | 33.57 | **16.26** | **39.26** | **21.82** | **39.90** | **18.62** | **48.70** | **36.92** | **34.69** | **17.67** |

### C.4 ABLATION STUDY OF META-OBJECTIVES

Our meta-objective consists of adversarial loss and self-supervised loss which are TRADES (Zhang et al., 2019) and the cosine similarity loss between differently augmented features as described in Equation 7. We compare two different meta-objectives of TROBA in the main paper, which are the case of using cross-entropy loss on clean examples instead of adversarial loss, and using only TRADES loss without self-supervised loss terms, respectively. Further, we replace the adversarial loss term with AT (Madry et al., 2018), which is widely used to obtain robustness in adversarial learning, while utilizing the same self-supervised loss. As shown in Table 8, utilizing a TRADES (Zhang et al., 2019) loss as an adversarial loss is more effective to obtain transferable robustness in adversarial meta-learning than a AT (Madry et al., 2018) loss.

### C.5 COMPARISON WITH SELF-SUPERVISED PRE-TRAINED MODELS

We select baseline models with ACL (Jiang et al., 2020)[3], BYORL (Gowal et al., 2020) and RoCL (Kim et al., 2020)[4] for self-supervised pre-trained baselines. We implement BYORL on top of the BYOL (Grill et al., 2020)[5] framework, following description in the paper.

## D ADDITIONAL EXPERIMENTAL RESULTS OF ROBUSTNESS

### D.1 ROBUSTNESS ON UNSEEN DOMAINS WITH DIFFERENT META-LEARNING FRAMEWORK AND DIFFERENT ITERATIONS OF BILEVEL ATTACK

To prove that TROBA is an effective method to obtain transferable robust representations, we experiment with three different types of meta-learning frameworks and different strengths of bilevel attacks. Specifically, we train TROBA on top of the MAML (Finn et al., 2017), FOMAML (Finn et al., 2017) and MetaSGD (Li et al., 2017) and apply bilevel attacks with 3 steps and 7 steps, respectively. Here, we only update the encoder parameters in inner adaption, since we propose task adaptive attacks that maximize the difference between the features, further to learn generalized representations as BOIL (Oh et al., 2020) demonstrated.

As shown in Table 9, TROBA outperforms the previous adversarial meta-learning model (Goldblum et al., 2020) by more than 10% robustness regardless of meta-learning strategies. Furthermore, we show outstanding robustness with only 3 steps of bilevel attacks (i.e., dynamic instance-wise attack) compared to AQ (Goldblum et al., 2020), which is trained with PGD-7 attacks (i.e., class-wise attack). To demonstrate that a bilevel attack is a more effective attack than a class-wise attack in the representation level, we calculate CKA (Kornblith et al., 2019) between clean and adversarial features to measure the similarity in the feature level. Notably, the CKA value of features attacked with the bilevel attack is smaller than the CKA values of features attacked with the class-wise attack (Figure 7), which means that the bilevel attack constructs more confusing perturbed images that are more dissimilar from their clean examples. Through these remarkable results, we demonstrate that our proposed bilevel attack served as a stronger attack that makes the model to have robust transferability to unseen domains, even with fewer gradient steps of attacks and little data.

### D.2 ROBUSTNESS ON UNSEEN DOMAINS WITH LARGER DATASETS

---

[3] https://github.com/VITA-Group/Adversarial-Contrastive-Learning
[4] https://github.com/Kim-Minseon/RoCLforself-supervisedlearning
[5] https://github.com/lucidrains/byol-pytorch

Table 9: Results of transferable robustness with different meta-learning framework and attack iteration in 5-shot tasks. All models are trained with 5-way 5-shot images on CIFAR-FS and Mini-ImageNet. Clean stands for test accuracy(%) of clean images. Rob. stands for accuracy(%) that is calculated with PGD-20 attack ($\epsilon = 8./255.$). All models are trained on ResNet12. The number of attack iteration during training is marked in parentheses next to the meta-train dataset. Further, we denote ($\theta$) next to the meta-learning strategies to notice that we update only the encoder parameters during inner optimization.

| | CIFAR-FS (3 steps) → | Mini-ImageNet | | tiered-ImageNet | | CUB | | Flowers | | Cars | |
|---|---|---|---|---|---|---|---|---|---|---|---|
| | | Clean | Rob. | Clean | Rob. | Clean | Rob. | Clean | Rob. | Clean | Rob. |
| TROBA | +MAML ($\theta$) (Finn et al., 2017) | 34.35 | 15.76 | 39.06 | 20.08 | 42.32 | 17.46 | 57.74 | 32.70 | 35.78 | 15.79 |
| | +FOMAML ($\theta$) (Finn et al., 2017) | 32.06 | 16.69 | 37.97 | 22.15 | 37.65 | 17.50 | 56.68 | 34.08 | 36.33 | 18.45 |
| | +MetaSGD ($\theta$) (Li et al., 2017) | 44.64 | 15.75 | 53.25 | 28.05 | 50.78 | 22.44 | 70.08 | 41.52 | 40.08 | 16.88 |
| | AQ (Goldblum et al., 2020) | 33.79 | 1.59 | 36.41 | 2.27 | 39.35 | 2.88 | 58.69 | 6.59 | 37.39 | 2.30 |
| | CIFAR-FS (7 steps) → | Mini-ImageNet | | tiered-ImageNet | | CUB | | Flowers | | Cars | |
| | | Clean | Rob. | Clean | Rob. | Clean | Rob. | Clean | Rob. | Clean | Rob. |
| TROBA | +MAML ($\theta$) (Finn et al., 2017) | 32.57 | 16.12 | 38.90 | 22.51 | 39.44 | 16.52 | 56.79 | 32.83 | 36.58 | 16.56 |
| | +FOMAML ($\theta$) (Finn et al., 2017) | 31.71 | 17.40 | 37.33 | 23.28 | 38.63 | 18.79 | 59.57 | 36.79 | 37.94 | 21.34 |
| | +MetaSGD ($\theta$) (Li et al., 2017) | 45.82 | 24.12 | 51.46 | 30.06 | 48.56 | 25.23 | 66.49 | 42.16 | 38.29 | 19.43 |
| | AQ (Goldblum et al., 2020) | 33.09 | 3.32 | 37.41 | 5.05 | 38.37 | 4.10 | 60.14 | 11.03 | 36.83 | 4.47 |
| | Mini-ImageNet (3 steps) → | CIFAR-FS | | tiered-ImageNet | | CUB | | Flowers | | Cars | |
| | | Clean | Rob. | Clean | Rob. | Clean | Rob. | Clean | Rob. | Clean | Rob. |
| TROBA | +MAML ($\theta$) (Finn et al., 2017) | 57.11 | 30.76 | 43.15 | 20.44 | 46.00 | 17.03 | 62.23 | 32.60 | 39.70 | 16.83 |
| | +FOMAML ($\theta$) (Finn et al., 2017) | 51.48 | 29.05 | 39.22 | 20.92 | 37.76 | 14.66 | 49.80 | 25.04 | 38.02 | 16.07 |
| | +MetaSGD ($\theta$) (Li et al., 2017) | 66.48 | 37.36 | 59.73 | 29.35 | 53.33 | 20.20 | 68.93 | 33.39 | 42.09 | 13.73 |
| | AQ (Goldblum et al., 2020) | 66.52 | 23.01 | 48.33 | 5.70 | 47.12 | 7.37 | 66.80 | 13.65 | 37.32 | 4.34 |
| | Mini-ImageNet (7 steps) → | CIFAR-FS | | tiered-ImageNet | | CUB | | Flowers | | Cars | |
| | | Clean | Rob. | Clean | Rob. | Clean | Rob. | Clean | Rob. | Clean | Rob. |
| TROBA | +MAML ($\theta$) (Finn et al., 2017) | 56.61 | 35.18 | 41.96 | 24.11 | 44.97 | 19.64 | 62.34 | 34.73 | 39.85 | 19.26 |
| | +FOMAML ($\theta$) (Finn et al., 2017) | 53.42 | 35.95 | 37.91 | 22.15 | 39.88 | 17.40 | 59.66 | 33.64 | 39.93 | 17.94 |
| | +MetaSGD ($\theta$) (Li et al., 2017) | 65.45 | 36.51 | 59.64 | 29.73 | 53.70 | 20.64 | 69.84 | 36.49 | 42.25 | 14.42 |
| | AQ (Goldblum et al., 2020) | 61.97 | 30.73 | 47.61 | 14.21 | 45.64 | 13.19 | 65.40 | 25.01 | 37.29 | 8.85 |

Table 10: Experiments results for self-supervised robust full-finetuning of TROBA and the state-of-the-art adversarial self-supervised models on unseen domains. While TROBA is trained on CIFAR-FS with bilevel attacks , adversarial self-supervised models are trained on full-dataset of CIFAR-100. All models are trained on ResNet18, and evaluated against PGD-20 attacks ($\epsilon = 8./255.$) and AutoAttack (AA) (Croce & Hein, 2020)

| | | CARS | | | CUB | | | AirCraft | | |
|---|---|---|---|---|---|---|---|---|---|---|
| | Method | Clean | PGD $\ell_\infty$ | AA | Clean | PGD $\ell_\infty$ | AA | Clean | PGD $\ell_\infty$ | AA |
| SSL | RoCL (Kim et al., 2020) | 35.00 | 8.11 | 5.67 | 17.21 | 2.55 | 1.71 | 33.63 | 8.76 | 5.61 |
| | ACL (Jiang et al., 2020) | 30.95 | 5.86 | 3.80 | 17.00 | 2.33 | 1.54 | 31.19 | 7.26 | 4.68 |
| | BYORL (Gowal et al., 2020) | 32.13 | 6.15 | 4.39 | 16.78 | 2.28 | 1.48 | 31.16 | 6.63 | 4.17 |
| | **Ours (3 steps)** | 31.47 | **9.58** | **6.19** | 18.07 | **4.49** | **2.73** | 32.12 | **9.93** | **6.19** |

In the main paper, we validate our models on unseen domains with larger benchmark datasets for standard image classification, which are CIFAR-10 and STL-10. Furthermore, we also demonstrate the robust transferability of our models in benchmark few-shot image classification tasks, which are Cars, CUB, and Aircraft that have 196, 200, and 100 classes, respectively. Especially, we train our models on ResNet18 with bilevel attacks with 3 steps while other self-supervised models are trained with PGD-7 attacks due to computation

Table 11: Test accuracy(%) of TROBA and self-supervised pre-traiend models on common corruption tasks of CIFAR-10.

| Model | Accuracy |
|---|---|
| ACL (Jiang et al., 2020) | 68.6 |
| BYORL (Gowal et al., 2020) | 69.01 |
| AQ (Goldblum et al., 2020) | 66.16 |
| TROBA | 67.9 |

costs. We use the same hyperparameters to validate with robust full-finetuning for all datasets, as we explained in Appendix B.5. Although our models utilize only scarce data to train, and even apply bilevel attacks with fewer gradient steps, we show even better robust representations compared to self-supervised pre-trained models while preserving clean accuracy (Table 10). Especially, our methods show a larger gap in fine-grained datasets, which have highly different distribution from meta-trained domains (i.e., CIFAR-FS). Further, we hope that our models to be robust in real-world adversarial perturbation such as common corruption (Hendrycks & Dietterich, 2019), we evaluate our fully finetuned models with adversarial examples on CIFAR-10, with common corruption datasets on

Table 12: Experiments results for robust full-finetuning of TROBA with and without self-supervised loss. Both models are trained on CIFAR-FS, with ResNet18 as the base encoder. Robust accuracy is evaluated with $l_\infty$ PGD-20 attacks with $\epsilon = 8./255.$ and the step size $\alpha = 8./2550.$, while AA stands for AutoAttack (Croce & Hein, 2020).

| | STL-10 | | | CARS | | | CUB | | | AirCraft | | |
|---|---|---|---|---|---|---|---|---|---|---|---|---|
| Loss | Clean | PGD $\ell_\infty$ | AA | Clean | PGD $\ell_\infty$ | AA | Clean | PGD $\ell_\infty$ | AA | Clean | PGD $\ell_\infty$ | AA |
| TRADES | **60.41** | 32.10 | 27.47 | 35.00 | 8.66 | 6.04 | 17.12 | 2.88 | 1.95 | 31.79 | 8.07 | 5.22 |
| TRADES+Self-sup | 53.46 | **32.65** | **28.96** | **31.47** | **9.58** | **6.19** | **18.07** | **4.49** | **2.73** | **32.12** | **9.93** | **6.48** |

Table 13: Results of transferable robustness in 5-shots unseen domain tasks. TROBA is trained on CIFAR-FS, with ResNet18 as the base encoder. On top of the pre-trained ROCL, we meta-train the model for 300 epochs as the same environment as TROBA is trained. Rob. stands for accuracy(%) that is calculated with PGD-20 attack ($\epsilon = 8./255.$, step size$=\epsilon/10$). Clean stands for test accuracy(%) of clean images.

| | mini-ImageNet | | Tiered-ImageNet | | CUB | | Flowers | | Cars | |
|---|---|---|---|---|---|---|---|---|---|---|
| | Clean | Rob. | Clean | Rob. | Clean | Rob. | Clean | Rob. | Clean | Rob. |
| TROBA | 33.57 | 16.26 | 39.26 | 21.82 | 39.90 | 18.62 | 48.70 | 36.92 | 34.69 | 17.67 |
| ROCL (Kim et al., 2020)+TROBA | 40.87 | 21.35 | 45.68 | 26.47 | 39.74 | 20.15 | 32.25 | 22.12 | 28.07 | 15.80 |

CIFAR-10. TROBA shows comparable accuracy with self-supervised pre-trained models on common corruption tasks, even trained with little data and bilevel attacks with fewer inner maximization iterations (Table 11). From these results, we prove that TROBA learns good generalized representations with little data effectively.

# E    EFFECT OF SELF-SUPERVISED CONCEPT IN TROBA

## E.1    SELF-SUPERVISED LOSS

Existing adversarial meta-learning works are vulnerable to unseen domains, since the regularization to learn good representations themselves is insufficient but normally focuses on optimization to rapidly adapt to new tasks. To learn good representations, we leverage self-supervised learning in adversarial meta-learning. Specifically, we enforce the consistent features between bilevel attacked images by using cosine similarity loss as a regularization. In robust full-finetuning, where larger data is used to finetune the model parameters, we observe that the model trained with self-supervised loss mostly shows better clean accuracy and robust accuracy on unseen domains (Table 12). Especially, self-supervised loss helps in obtaining generalized features more as the distribution of the target domains is more different from the distribution of the meta-trained dataset.

## E.2    TROBA ON TOP OF THE PRE-TRAINED SELF-SUPERVISED MODEL

We further verify whether our methods can support self-supervised pre-trained models to obtain transferable robustness in few-shot tasks on unseen domains. Especially, we utilize ROCL (Kim et al., 2020) pre-trained on CIFAR-100, and further meta-train with TROBA on top of that model. As shown in Table 13, since self-supervised models learn representations that are color-invariant where *colorjitter* is normally used in data augmentation, TROBA is less effective in obtaining robust representations in the fine-grained datasets (i.e., CUB, Flowers, Cars) that are easily affected by colors. On the other hand, the meta-trained self-supervised pre-trained model achieves even better clean accuracy and robustness than TROBA, especially in general datasets (i.e, Mini-ImageNet, Tiered-ImageNet), where TROBA helps in generalization on few-shot tasks on general domains.

# F    OBFUSCATED GRADIENT

All of the robust accuracies in our paper are calculated with the strength $\epsilon = 8./255.$, step size $\alpha = 8./2550.$ and 20 steps. To check whether our model is under obfuscated gradient issues or not, we experiment with two different settings of $l_\infty$ PGD attacks. First, we apply PGD attacks with extremely large strength, where robust accuracy should be almost zero. Second, we use the same strength but different step sizes and steps, which are $4./2550.$ and $40$, respectively, where robust

Table 14: Test accuracy(%) on benchmark data sets for 5-shots. Robustness is calculated with PGD-20 attack ($\epsilon = 8./255.$, step size=$\epsilon/10$), clean is for clean images. All models are adversarially meta-trained on CIFAR-FS.

| | Strength ($\epsilon$) | Step size ($\alpha$) | Steps | CIFAR-FS | | Mini-ImageNet | | Tiered-ImageNet | | CUB | | Cars | |
|---|---|---|---|---|---|---|---|---|---|---|---|---|---|
| | | | | Clean | PGD $\ell_\infty$ | Clean | PGD $\ell_\infty$ | Clean | PGD $\ell_\infty$ | Clean | PGD $\ell_\infty$ | Clean | PGD $\ell_\infty$ |
| 3 steps | 8.0/255.0 | 8.0/2550.0 | 20 | 53.42 | 35.95 | 32.06 | 16.69 | 37.97 | 22.15 | 37.65 | 17.50 | 36.33 | 18.45 |
| | 8.0/255.0 | 4.0/2550.0 | 40 | 53.04 | 35.35 | 31.70 | 16.01 | 38.06 | 21.98 | 37.77 | 18.12 | 36.10 | 18.02 |
| | 300.0 | 8.0/2550.0 | 20 | 52.72 | 0.47 | 31.83 | 0.92 | 37.73 | 0.85 | 38.14 | 0.55 | 36.21 | 0.44 |
| 7 steps | 8.0/255.0 | 8.0/2550.0 | 20 | 51.90 | 36.01 | 31.71 | 17.40 | 37.33 | 23.28 | 38.63 | 18.79 | 37.94 | 21.34 |
| | 8.0/255.0 | 4.0/2550.0 | 40 | 52.50 | 36.39 | 31.95 | 17.49 | 38.44 | 24.22 | 38.18 | 18.87 | 37.41 | 20.92 |
| | 300.0 | 8.0/2550.0 | 20 | 52.20 | 0.50 | 31.97 | 0.59 | 37.53 | 0.65 | 38.78 | 0.45 | 37.64 | 0.48 |

accuracy should be the same as robust accuracy from our original evaluation setting. Specifically, we demonstrate TROBA trained on CIFAR-FS with ResNet12 as the base encoder, and further on top of the FOMAML reported in Table 9. As shown in Table 14, we verify that our models do not have any obfuscated gradient issues.

## G VISUALIZATION OF LOSS SURFACE

We visualize the loss surface of our model and baseline AQ (Goldblum et al., 2020) model. As shown in the Figure our model has a smoother loss surface both in the seen domain and unseen domain while the baseline has a relatively less smooth surface.

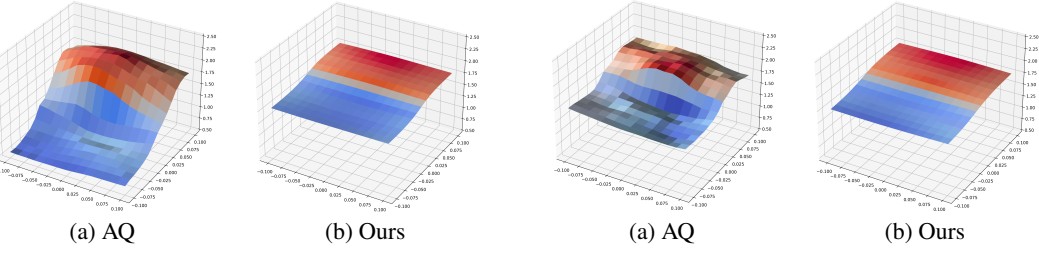

(a) AQ  (b) Ours  (a) AQ  (b) Ours

Figure 9: Seen domain (CIFAR-FS)  Figure 10: Unseen domain (Mini-ImageNet)

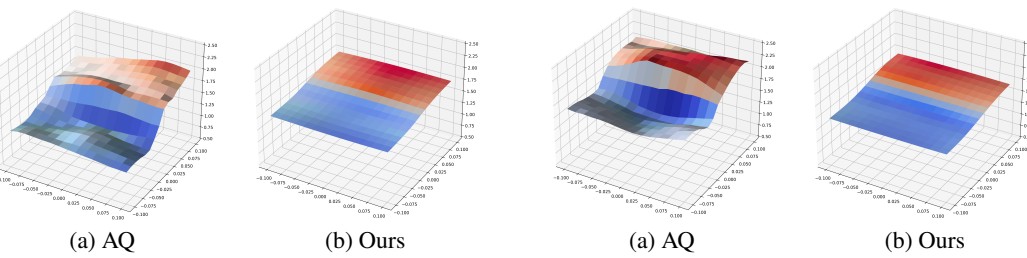

(a) AQ  (b) Ours  (a) AQ  (b) Ours

Figure 11: Unseen domain (Tiered-ImageNet)  Figure 12: Unseen domain (CUB)

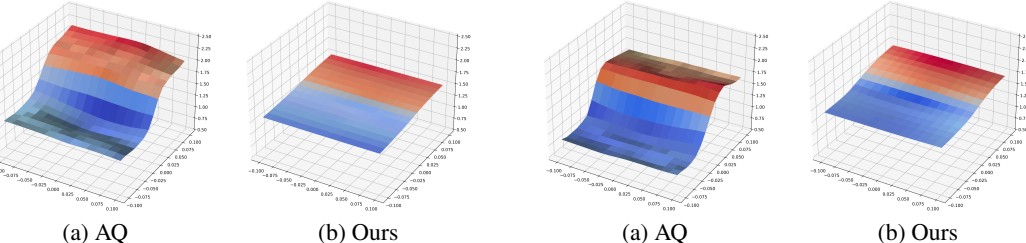

(a) AQ  (b) Ours  (a) AQ  (b) Ours

Figure 13: Unseen domain (Cars)  Figure 14: Unseen domain (Flowers)

