# OpenReview forum: "Few-Shot Transferable Robust Representation Learning via Bilevel Attacks"
_ICLR.cc/2023/Conference — Submitted to ICLR 2023_

### Official Review · Reviewer_WWPz · 2022-10-13

**Confidence:** 3
**Correctness:** 3
**Technical Novelty And Significance:** 3
**Empirical Novelty And Significance:** 3
**Recommendation:** 6

**Clarity, Quality, Novelty And Reproducibility:**

The writing is generally clear, although like I indicated above, the notation in Algorithm 1 is hard to understand.

**Strength And Weaknesses:**

In my opinion, the strength here is the transferability of robustness across domains.  The empirical results otherwise do not seem very strong.

It would be good to test on higher performing meta-learning algorithms such as R2D2 and MetaOptNet, as in the AQ paper, at least as baselines for AQ since they may not be applicable to your method instead of only testing on MAML.  The numbers in the AQ paper look higher even for directly comparable experiments such as seen domain 5-shot mini-ImageNet.  It seems like a major flaw that your method can only be used with MAML despite MAML not being a very good few-shot learning algorithm.  This is my central critique of this paper.

Algorithm 1 is impossible to understand, especially the arguments of the loss function in the gradient computation.

Some of the tables are a bit weird and contain numbers which are very hard to compare.  For example, Table 7 contains the proposed method as well as other methods trained against other stronger attacks.  In many cases, the competitors outperform the proposed method.


**Summary Of The Paper:**

This paper develops a new algorithm for adversarially robust few-shot MAML.  The interesting part is that the adversarial robustness transfers across domains.  Experiments are conducted on multiple datasets and against multiple baseline methods.

**Summary Of The Review:**

This area of work is valuable, but the proposed method seems only applicable to MAML even though the baseline methods work with better quality few-shot learning algorithms.  This might be a major performance limitation of this paper.  In order to convince me that it is not a limitation, I would suggest either adapting the method to a better few-shot learner than MAML or demonstrate that it can outcompete competitors even when they use very strong meta-learning algorithms.

---

> ### Author Response · Authors · 2022-11-11
> **Response to your comments (2/2)**
>
> **[Algorithm 1 is difficult to understand]**
> > Algorithm 1 is impossible to understand, especially the arguments of the loss function in the gradient computation.
> - Due to the page limit, we used abbreviations in Algorithm 1 while explaining the detail of the loss function in Eq.(7) in the main text. However, reflecting on your comments, we rewrote Algorithm 1 (blue text) to improve its clarity. To be more precise, we independently compute the gradient in each encoder which is parameter-level augmented. Therefore, the sum of *our loss* function described in Eq. (7) of three images, $T_1(x)$, $T_2(x)$, $T_1(x)^{adv}$, are computed on $\theta_1$, and the sum of *our loss* function of three images, $T_1(x)$, $T_2(x)$, $T_2(x)^{adv}$, are computed on $\theta_2$.
>
> **[Table is hard to compare]**
> > Some of the tables are a bit weird and contain numbers which are very hard to compare. For example, Table 7 contains the proposed method as well as other methods trained against other stronger attacks. In many cases, the competitors outperform the proposed method.
> - We will clarify the organization of Table 7. Please note that Table 7 does not serve the purpose of showing the superiority of our method to self-supervised learning, since the comparison is unfair to ours as SSL is trained with a significantly larger amount of training data. The results in Table 7, on the other hand, show that our model achieves comparable robustness **even with a smaller training set** and **weaker attack strengths** during the training compared to adversarial self-supervised learning approaches trained with a larger amount of data. This is somewhat surprising and shows the clear effectiveness of our method.
> - Furthermore, our methods highly outperform the competitors in unseen domain tasks (Please see Table 1, Table 2).

---

> > ### Comment · Reviewer_WWPz · 2022-11-25
> > **Thanks for your response**
> >
> > I have increased my score to 6 as the authors have in part addressed my feedback.
> >
> > Your rebuttal indicates that your model may have poor performance because of the small image resolution used for the purposes of transferring to other domains with small resolutions, yet, you could easily upsample images with small resolution just as people do when they transfer from ImageNet to datasets with small image resolutions.  I understand that previous works use 32x32 too, but it would be a major limitation if your in-domain accuracy has to be this poor.  It also makes me question if there are better ways to learn good models with higher resolution images on the upstream tasks and get better performance on the low-resolution downstream tasks too.
> >
> > Also, “does not serve the purpose of showing the superiority of our method to self-supervised learning” is incongruous with the statement in the abstract that the proposed method significantly outperforms SSL.

---

> > > ### Author Response · Authors · 2022-11-25
> > > **Thank you for your valuable comments**
> > >
> > > Dear reviewer,
> > >
> > > We sincerely appreciate your suggestions and insightful comments.
> > >
> > > We will reflect your suggestion and proceed to train our models on higher resolution images (i.e., 84x84 for Mini-ImageNet) during the remaining rebuttal period or until camera-ready, which we believe will further strengthen our paper. Moreover, we will also check whether our model could obtain better performances on low-resolution downstream tasks when TROBA is trained on upstream tasks with higher resolutions.
> > >
> > > We will also revise the statement in the abstract that our model significantly outperforms SSL, and tone down the statement to "our model could obtain effective transferrable representations for cross-domain tasks that are comparable to the performance of SSL, with few training instances."
> > >
> > > Thank you again for your effort and time to review our paper, and your feedback during the rebuttal. We really appreciate your help.
> > >
> > > Sincerely, \
> > > Authors

---

> ### Author Response · Authors · 2022-11-11
> **Response to your comments (1/2)**
>
> **[The empirical results otherwise do not seem very strong.]**
> > In my opinion, the strength here is the transferability of robustness across domains. The empirical results otherwise do not seem very strong.
> It would be good to test on higher performing meta-learning algorithms such as R2D2 and MetaOptNet, as in the AQ paper, at least as baselines for AQ since they may not be applicable to your method instead of only testing on MAML. The numbers in the AQ paper look higher even for directly comparable experiments such as seen domain 5-shot mini-ImageNet. It seems like a major flaw that your method can only be used with MAML despite MAML not being a very good few-shot learning algorithm. This is my central critique of this paper.
> - First of all, bilevel parameter augmentation is applicable to gradient-based bilevel optimization meta-learning approaches (See Table 4.) and we mainly employ MetaSGD on our TROBA. Further, we report all of the performance of **AQ trained on R2D2**, not on MAML. We just train AQ on the **official codes** with only re-implementing backbone architecture from R2D2 to ResNet12 while the entire meta-learning framework is maintained as R2D2 for a fair comparison. In the case of Mini-ImageNet, the performance gap comes from **difference in the image size** used in training. AQ utilizes an image size of 84 for MiniImageNet since they do not evaluate the model on unseen domains while we utilize an image size of 32 for all datasets to show transferability on unseen domains as followed by previous cross-domain meta-learning works [Oh et al., Ryu et al, R Volpi et al.].
> - As you commented we test AQ with an R2D2 backbone and R2D2 meta-learning framework. **Our TROBA still shows about 5%~10% better robustness in the cross-domain tasks even compared to AQ (R2D2) models.** From these results, we strongly believe our empirical results show the effectiveness of our TROBA compared to previous works.
> | method |  architecture of backbone |  meta-learning framework  | CIFAR-FS | 	     | Mini-ImageNet  |           | Tiered-ImageNet  |           |  CUB  |           |
> |----------------|-----------|-----------|-----------|-----------|-----------|-----------|---------------|-----------|-----------|-----------|
> |          | |      | Clean     | PGD       | Clean     | PGD       | Clean         | PGD       | Clean     | PGD       |
> | AQ | R2D2 | R2D2 | 73.19 | 42.82  | 36.46 | 16.10  |  41.89 | 21.78 | **50.97**  |20.78 |
> | AQ | ResNet12 | R2D2 | **73.49** | 28.49  | 33.09 | 3.32  | 37.41 | 5.05 | 38.37  | 4.10 |
> | TROBA | ResNet12 | MetaSGD | 64.90 | **43.34** | **45.82** | **24.12** | **51.46** | **30.06**  | 48.56 | **25.23** |
>
> [Ryu et al.] MetaPerturb: Transferable Regularizer for Heterogeneous Tasks and Architectures, NeurIPS 2020 \
> [R Volpi et al.] Continual Adaptation of Visual Representations via Domain Randomization and Meta-learning, CVPR, 2021 \
> [Oh et al.] BOIL: TOWARDS REPRESENTATION CHANGE FOR FEW-SHOT LEARNING, ICLR 2021

---

> ### Author Response · Authors · 2022-11-25
> **A Gentle Reminder**
>
> Dear reviewer WWPz,
>
> Initially, you raised concerns about our baselines. Therefore, we reimplemented the baseline **AQ with a stronger backbone and framework (R2D2)** to compare its performance against ours. We found that **our work still achieves significantly superior performance to the stronger AQ baseline on our main experiments, which is to obtain robustness on unseen domains**.
>
> We believe that we have successfully addressed your concern on this matter, and that addressing this point has further strengthened our work. Thank you for your constructive comments.
>
> Best,\
> Authors

---

### Official Review · Reviewer_Xygn · 2022-10-25

**Confidence:** 3
**Correctness:** 3
**Technical Novelty And Significance:** 2
**Empirical Novelty And Significance:** 3
**Recommendation:** 6

**Clarity, Quality, Novelty And Reproducibility:**

see the strength and weaknesses. It provides detailed instructions about how to reproduce the results in figures and tables in the appendix. It  It provides visualization results to demonstrate why it performs better, which is easy to follow. The technical contribution may be limited as it adopts many existing techniques.

**Strength And Weaknesses:**

\+ The visualization of the representations of the instance-wise adversarial samples from the unseen domains in Figure 2 shows that the model is able to obtain well-separated feature space for attacked samples on this domain (CIFAR-10) even before adapting to it, while
the previous adversarial meta-learning framework learns a feature space with large overlaps across the adversarial instances belonging
to different classes.

\+ TROBA shows smoother loss surface to adversarial examples in Figure 4 which is directly associated with better robustness and generalization.

\+ The ablation study checks the performance of each component, such as it compares the performance with or without augmentation, and with instance-wise or class-wise attack. It also checks the performance of the loss function compared with CE + selfsup loss or TRADES loss only.

\- The technical contribution may be limited. It adopts many techniques in this paper. Most of techniques are proposed in prior works such as
TRADE loss,  data augmentation with random crop/flip/color distortion, and instance-wise classification loss in adversarial self-supervised learning. It combines these techniques but the technical contribution may not be significant.

**Summary Of The Paper:**

It proposes a adversarial meta-learning framework with bilevel attacks, which allows the model to learn generalizable robust representations across tasks and domains. The framework obtains higher robustness in few-shot tasks both in the seen domain and the unseen domains.

**Summary Of The Review:**

It combines a few existing techniques to achieve good performance in adversarial meta-learning. It provides visualization results to demonstrate why it performs better, which is easy to follow. My main concern is that the technical contribution may be limited as most of the techniques are adopted from prior works.

---

> ### Author Response · Authors · 2022-11-11
> **Response to your comments**
>
> Thank you for your comments and highlighted strength of our works. We response to your concerns of our work.
>
> **[Technical contribution may be limited. It adopts many existing techniques.]**
> >The technical contribution may be limited. It adopts many techniques in this paper. Most of techniques are proposed in prior works such as TRADE loss, data augmentation with random crop/flip/color distortion, and instance-wise classification loss in adversarial self-supervised learning. It combines these techniques but the technical contribution may not be significant.
> * Please note that the novelty of our work is in the proposal of a novel "adversarial meta-learning framework". Thus the novelty is in the framework itself which allows to perform inner gradient steps on the parameter of the encoders for two different views with differently augmented sets of samples, to maximize the disagreement across the views.
> * Our **bilevel parameter augmentation** is novel in the perspective, since none of the existing self-supervised learning methods **aim to generate the disagreement across the views by updating the encoders, while minimizing the disagreement at the meta-level.**
> * We propose bilevel augmentation which is our novel and also **core component to link instance-wise attack and self-supervised loss to meta-learning framework into TROBA** which is the **first attempt to employ the techniques in diverse domain to adversarial meta-learning context.**
> * Moreover, we do not aim toward achieving robustness to unseen tasks from the same domain, but to achieve robustness to unseen domains. Thus the problem our tackle is completely different from that of AQ and we also have a contribution in terms of the problem we tackle. As highlighted by Reviewer WWPz, KTHv, we believe that a robust meta-learning framework across the unseen domains is an important problem, yet an under-explored and non trivial problem to solve, and our firstly proposed TROBA could be an important milestone in this direction. In this regard, we believe our key contributions can also be found in the unexplored problem itself.
> * Also, our framework is not a trivial combination of adversarial learning, meta-learning and self-supervised learning. Please see the results in Table 6, where such a simple combination cannot achieve same performance as our TROBA.

---

> ### Author Response · Authors · 2022-11-25
> **A Gentle Reminder**
>
> Dear reviewer Xygn,
>
> We addressed your concerns on the technical novelty of our work. In the revised version of the paper, we clearly elaborated our contribution of **bilevel parameter augmentation for meta-adversarial learning** which was not introduced by any existing works on adversarial training, self-supervised learning, or meta-learning. The bilevel parameter augmentation is a core mechanism to link three different domain techniques to solve **a novel problem of achieving robustness in unseen domains**. Further, we also experimentally demonstrated that **a simple combination of previous works could not obtain the same performance as our TROBA**.  Please find our response below and in the revision, and let us know if there are any remaining concerns.
>
> |                | CIFAR-FS  |           | Tiered ImageNet   |           | Cars  |           |
> |----------------|-----------|-----------|---------------|-----------|-----------|-----------|
> |                | Clean     | PGD       | Clean         | PGD       | Clean     | PGD       |
> | Single encoder instance-wise attack | 63.10     | 36.98     | 51.57     | 25.05     | 38.99 | 14.36 |
> | Bilevel Aug class-wise attack | **66.69**     | 40.48    | **53.91**  | 27.41     | 69.66 | 38.83 | 40.05 | 16.37 |
> | TROBA (Bilevel Aug  instance-wise attack)  | 65.82 | **41.39** | 53.25 | **28.05** | **40.08** | **16.88** |
>
> We would be more than happy to address your concerns.
>
> Best, \
> Authors

---

### Official Review · Reviewer_KTHv · 2022-10-25

**Confidence:** 3
**Correctness:** 3
**Technical Novelty And Significance:** 1
**Empirical Novelty And Significance:** 2
**Recommendation:** 5

**Clarity, Quality, Novelty And Reproducibility:**

Most explanations of the proposed method are clear and are presented with high quality. The technique is not novel and some implementation details are not given.

**Strength And Weaknesses:**

Strength:

The studied problem is interesting and practical as learning with few examples per task is important in some applications.

Experimental results show the effectiveness of the proposed loss function in terms of robustness. Ablation studies also verify that each individual design is effective.


Weaknesses:

The major weakness of this work is that the proposed loss function seems to be a combination of the adversarial meta-learning method AQ and the objective function of contrastive adversarial learning. The proposed loss function thus inherited the advantages from both domains. The technique is not novel.

The title of the paper suggests learning transferable representations for adversarial few-shot learning. My feeling is that the word transferable seems to be redundant since meta-learning for few-shot learning itself already has the implication of improving the transferability of representations. This work mainly uses self-supervised learning methods, specifically contrastive learning loss function, to learn better representations that can transfer to different tasks or domains without explicit transfer methods (e.g. things like task augmentation or domain alignment) beside meta-learning. Therefore, in my opinion it is perhaps better to claim it as few-shot self-supervised robust representation learning.


**Summary Of The Paper:**

This paper proposes a method to solve the problem of adversarial meta-learning, where the difficulty lies in the few-shot learning of each inner loop task. The proposed method consists of 1) generating adversarial query examples by attacking a contrastive loss between clean and perturbed images; 2) using two independent augmentations in both inner loop optimization and attack; 3) adversarial meta learning with TRADES loss and a contrastive loss. Experimental results show that the proposed method achieves improved robustness performance compared with several baselines.

**Summary Of The Review:**

The proposed method targets the problem of adversarial meta-learning using techniques from contrastive learning. The incorporation of existing loss functions is not novel, and performance gain in experiments are expected due to the effectiveness of contrastive learning loss functions.

---

> ### Author Response · Authors · 2022-11-11
> **Response to your comments**
>
> Thank you for your comments. We will respond to your concerns.
>
> **[1. Technique is not novel. It is simple combination]**
> >The major weakness of this work is that the proposed loss function seems to be a combination of the adversarial meta-learning method AQ and the objective function of contrastive adversarial learning. The proposed loss function thus inherited the advantages from both domains. The technique is not novel.
>
> - Please note that the novelty of our work is in the proposal of a novel "adversarial meta-learning framework". Thus the novelty is in the framework itself which allows performing inner gradient steps on the parameter of the encoders for two different views with differently augmented sets of samples, to maximize the disagreement across the views.
>
> - Our **bilevel parameter augmentation** is novel in the perspective of self-supervised learning, since none of the existing self-supervised learning methods aim to **generate disagreement across the views by updating the encoders while minimizing the disagreement at the meta-level**.
>
> - Moreover, we do not aim toward achieving robustness to unseen tasks from the same domain, but to achieve **robustness to unseen domains**. Thus the problem we tackle is completely different from that of AQ and we also have a contribution in terms of the problem we tackle. As highlighted by Reviewer WWPz, we believe that a robust meta-learning framework across the unseen domains is an important problem, yet an under-explored and non-trivial problem to solve, and our firstly proposed TROBA could be an important milestone in this direction. In this regard, we believe our key contributions can also be found in the unexplored problem itself.
>
> - Also, our framework is not a trivial combination of adversarial meta-learning and contrastive learning. Please see the results in Table 6, where such a simple combination cannot achieve the same performance as our TROBA.
>
>
> **[2. Title should be changed]**
> >The title of the paper suggests learning transferable representations for adversarial few-shot learning. My feeling is that the word transferable seems to be redundant since meta-learning for few-shot learning itself already has the implication of improving the transferability of representations. This work mainly uses self-supervised learning methods, specifically contrastive learning loss function, to learn better representations that can transfer to different tasks or domains without explicit transfer methods (e.g. things like task augmentation or domain alignment) besides meta-learning. Therefore, in my opinion, it is perhaps better to claim it as few-shot self-supervised robust representation learning.
>
> - We do believe that meta-learning is the most crucial part of our work, although our method is inspired by self-supervised learning. Please note that although our work mainly gets motivation from self-supervised learning approaches, since our method uses **label information**, it is not appropriate to call it self-supervised learning, as self-supervised learning is an unsupervised learning problem.
>
> - Further, our main goal is to obtain a model with **transferable robustness to unseen domains with few data**, which is a meta-learning problem. Moreover, we introduce **bilevel attack** along with **bilevel parameter augmentation** that effectively utilizes the inner adaptation of meta-learning, which means meta-learning is the core part of our work. The experimental results demonstrate the importance of our bilevel attack compared to simple techniques (Please see Table 5), and thus we strongly believe that **bilevel attack, a meta-learning component**, is our main contribution and the essential component to describe our works.

---

> ### Author Response · Authors · 2022-11-25
> **A Gentle Reminder**
>
> Dear KTHv,
>
> You raised initial concerns that our work is a simple combination of previous works and title is redundant.
>
> However, this is a misunderstanding and our framework is not a simple combination of previous works. We also experimentally demonstrated that **such a simple combination of meta-learning and adversarial learning (single encoder instance-wise attack) cannot obtain the same performance as our TROBA**.
> |                | CIFAR-FS  |           | Tiered ImageNet   |           | Cars  |           |
> |----------------|-----------|-----------|---------------|-----------|-----------|-----------|
> |                | Clean     | PGD       | Clean         | PGD       | Clean     | PGD       |
> | Single encoder instance-wise attack | 63.10     | 36.98     | 51.57     | 25.05     | 38.99 | 14.36 |
> | Bilevel Aug class-wise attack | **66.69**     | 40.48    | **53.91**  | 27.41     | 69.66 | 38.83 | 40.05 | 16.37 |
> | TROBA (Bilevel Aug  instance-wise attack)  | 65.82 | **41.39** | 53.25 | **28.05** | **40.08** | **16.88** |
>
> (single encoder instance-wise attack: For comparison, we simply apply instance-wise attack on the AQ model instead of class-wise attack.)
>
> **Our bilevel parameter augmentation is novel** as adapting the encoder for each view to the augmented samples has never been done in any of the previous works, and this is our core component that enables to obtain robust representations for cross-domain tasks.
>
> We further revised our manuscript to address your concerns on the novelty. Please go over the updates and reflect them to your final review and the rating of our work. Please let us know if you have any remaining questions or concerns, since we will be more than happy to address them.
>
> Best,\
> Authors

---

> ### Author Response · Authors · 2022-11-26
> **A gentle reminder**
>
> Dear reviewer KTHv,
>
> Thank you for your comments, we believe that we have faithfully addressed all your concerns during the rebuttal period.
>
> You seem to consider our work as a simple combination of adversarial meta-learning (AQ) and the objective function of contrastive adversarial learning. However, this is a critical misunderstanding, since 1) we do not use contrastive loss but non-contrastive self-supervised learning loss based on TRADES, and 2) our framework is not a simple combination of previous methods.
>
> We proposed a **novel bilevel parameter augmentation scheme** which can generate more effective adversaries by taking inner gradient steps on the shared encoder parameters, by computing gradients on the instance-wise adversarial examples. Such dynamic update of the encoder leads to more effective adversarial examples generated with stronger attacks, which results in larger discrepancies among the augmented samples in two different views. Moreover, meta-learning of the shared encoder to minimize the self-supervised learning loss allows us to find a good representation that can generalize across domains.
>
> We hope that this explanation of the bi-level parameter augmentation scheme resolves your misunderstanding of the novelty of the work. Please let us know if there are any remaining concerns.
>
> Thanks,\
> Authors

---

> > ### Comment · Reviewer_KTHv · 2022-11-28
> > **Thanks for your response**
> >
> > Dear Authors,
> >
> > I would like to thank the authors for new clarification on the overall novelty and difference compared with several previous work. In terms of the technical contribution, I think the only new part is the proposed bilevel parameter augmentation, while I don’t think it is a very principled approach. Therefore, shared with other reviewers’ concerns, I think the technical novelty is only incremental. I have changed my score to reflect the above points.
> >
> > Thanks.

---

> > > ### Author Response · Authors · 2022-12-01
> > > **Technical contributions**
> > >
> > > Thank you for your response. We appreciate that you increased your score based on our response.
> > >
> > > However, we believe that introduction of the bilevel parameter augmentation scheme is a meaningful technical contribution in the context of adversarial self-supervised learning for robust, transferrable representation learning.
> > >
> > > We have formulated the problem of learning generalizable robust representations into **a single, unified optimization objective with clear motivations for both the inner and the outer optimization problems. Eq. 6 is actually a min-max optimization objective** whose inner maximization problem aims to maximize $L_\texttt{similarity}$ in Eq.5, as follows:
> > >
> > > $\underset{\theta_1,\theta_2, \mu}{\operatorname{argmin}}E_{Q(x^q, y^q)\sim\mathbb{D}}[\underset{\delta_1, \delta_2 \in B(\cdot, \epsilon)}{\operatorname{max}} \left(L_{\texttt{similarity}}(f(\theta_1, t_1(x^q)+\delta_1), f(\theta_1, t_2(x^q))) + L_{\texttt{similarity}}(f(\theta_2, t_1(x^q)), f(\theta_2, t_2(x^q)+\delta_2))\right)]$
> > >
> > > - In the inner maximization problem, we maximize the disagreement across views in the inner optimization objective (Eq. 5).
> > > - In the outer minimization problem, we learn shared parameters that can minimize the discrepancy between two views across domains at the outer optimization objective (Eq. 6).
> > >
> > > In conclusion, our novel bilevel parameter augmentation framework consists of **a clear optimization objective** rather than a set of heuristics to achieve the **goal of generalizable robust representation learning across domains**, and we thus believe that it is sufficiently principled.

---

### Official Review · Reviewer_NDZi · 2022-10-26

**Confidence:** 4
**Correctness:** 3
**Technical Novelty And Significance:** 3
**Empirical Novelty And Significance:** 2
**Recommendation:** 6

**Clarity, Quality, Novelty And Reproducibility:**

- **Clarity**: Good.
- **Quality**: Good.
- **Novelty**: Incremental.
- **Reproducibility**: Great.

**Strength And Weaknesses:**

- **Strength**:
1. This paper is well organized and easy to follow.
2. The core parts of the proposed TROBA are clear.
3. The experimental results seem to be good.

- **Weakness**:
Overall, the key contributions especially the technique contribution are not clear. This is because self-supervised adversarial learning and TRADES have been widely used in general adversarial training, and BOIL is one existing meta-learning method extended from MAML. If the bilevel parameter augmentation and attack are the main technique contribution, the difference and effectiveness of this strategy compared to the normal data augmentation used in general self-supervised adversarial learning should be clearly claimed and demonstrated. If the entire framework is the main contribution, it would be interesting to see the effectiveness of this framework of using different adversarial training methods and different meta-learning methods.

- Several main concerns are as follows:
1. What’s the main difference between the proposed bilevel parameter augmentation and the normal data augmentation? Is it different only because of the usage of Siamese network with different parameters? Could you verify the effectiveness of this operation by experiments?
2. What’s the main difference between the bilevel attack and the normal self-supervised adversarial learning? Is it different only because of the swapping operation of two views? Could you verify the effectiveness of this operation by experiments?
3. It is not clear if the adversarial example l_n^adv used in Eq. (7) is generated by class-wise attack or instance-wise attack.
4. It is not clear how to update the backbone and classifier in Eq. (2) for readers. Note that the classifier is fixed in BOIL.
5. I am not sure if all the comparison methods employ the same backbone and the same epsilon of attack. Does this paper re-implement all these methods in the same framework?
6. It is doubtful for the experiment in Table 7. Are the results still for 5-shot classification? Note that the splits used in CIFAR-FS and CIFAR-100 are different. Why not directly perform other models on CIFAR-FS? How many samples used for training for TROBA and how many samples used for other models? This part should be clearly explained, otherwise the results are doubtful.
7. Do all the comparison methods employ the same input image size of 32 * 32?


**Summary Of The Paper:**

This paper introduces self-supervised adversarial learning into adversarial meta-learning, and proposes a transferable robust meta-learning via bilevel attack (TROBA) method by using a bilevel attack scheme. Specifically, TROBA is built on BOIL and TRADES by adding a self-supervised loss. Extensive experiments effectively show the effectiveness of the proposed method.

**Summary Of The Review:**

The overall of this work is great. However, there are still multiple parts that are not clear, which need to be further clarified.

---

> ### Author Response · Authors · 2022-11-11
> **Response to your comments (1/2)**
>
> Thank you for your valuable comments. We respond to your concerns below.
>
> **[Overall weakness] The key contributions are not clear.**
> > The key contributions especially the technique contribution are not clear. If the bilevel parameter augmentation and attack are the main technique contribution, the difference and effectiveness of this strategy compared to the normal data augmentation used in general self-supervised adversarial learning should be clearly claimed and demonstrated. If the entire framework is the main contribution, it would be interesting to see the effectiveness of this framework in using different adversarial training methods and different meta-learning methods.
>
> - First of all, the key contribution of our technique is the proposal of **a novel adversarial meta-learning framework with bilevel attack** which consists of bilevel parameter augmentation and dynamic instance-wise attack. We compare our work with normal data augmentation in Table 5. We clearly claim and show that our bilevel parameter augmentation improves the robustness in the unseen domain. Further, we also analyze how our bilevel augmentation affects having better robust latent space in Figure 5 and smoother loss landscape in Figures 3,4, and 8~13.  Moreover, we also compare our framework of using different adversarial training methods in Table 8 and meta-learning frameworks in Table 4 and Table 9.
> - Furthermore, as highlighted by Reviewer WWPz, KTHv, we believe that the problem we tackle, **adversarial meta-learning framework across the unseen domains** is an important yet under-explored problem, and our proposed framework could be an important milestone in this direction as **our work is the first to tackle such a problem.**
>
> **1. [Main difference between bilevel parameter augmentation and normal data augmentation]**
> - Our bilevel parameter augmentation is the idea to augment the encoder parameter with the inner adaptation loop with the set of data-augmented samples which is distinctively defined in a meta-learning framework (We will clarify the model figure in the supplementary (blue text)). In more detail, as shown in Algorithm 1, bilevel parameter augmentation produces two differently updated encoder parameters $\theta_1^\tau$ and $\theta_2^\tau$ for differently transformed images $t_1(x^s)$ and $t_2(x^s)$, respectively. To the best of our knowledge, our bilevel parameter augmentation is a **novel concept that has not been introduced** in self-supervised learning, adversarial learning, or meta-learning. Further, we verify the effectiveness of this contribution by empirical performance in Table 5 (recap Table 5 in the following table) and analyze its effect in the latent space in Figure 5.
>
> |                | CIFAR-FS  |           | Mini-ImageNet |           | Tiered-ImageNet   |           |
> |----------------|-----------|-----------|---------------|-----------|-----------|-----------|
> |                | Clean     | PGD       | Clean         | PGD       | Clean     | PGD       |
> | Image Aug only | 63.10     | 36.98     | 39.54         | 15.08     | 51.57     | 25.05     |
> | Bilevel Aug    | **65.82** | **41.39** | **44.64**     | **15.75** | **53.25** | **28.05** |
>
> **2. [Main difference between bilevel attack and instance-wise attack in adversarial SSL]**
> - As we mentioned in Section 3.2, our main contribution in dynamic instance-wise attack is **redesigning an instance-wise attack on top of bilevel parameter augmentation** where we generate attack images with **differently augmented encoders that are adapted via inner gradient updates on the augmented samples**.
> - Although a bilevel attack could seemingly be a simple combination that applies instance-wise attack in the meta-learning framework, our approach can only be defined on top of the bilevel parameter augmentation. Further, a simple combination of the instance-wise attack with the meta-learning framework, cannot achieve the level of robustness obtained by our TROBA (See the table below).
>
> |                | CIFAR-FS  |           | Tiered-ImageNet   |           | Cars |           |
> |----------------|-----------|-----------|---------------|-----------|-----------|-----------|
> |                | Clean     | PGD       | Clean         | PGD       | Clean     | PGD       |
> | Single encoder instance-wise attack | 63.10     | 36.98     | 51.57     | 25.05     | 38.99 | 14.36 |
> | Bilevel Aug class-wise attack | **66.69**     | 40.48    | **53.91**  | 27.41     | 40.05 | 16.37 |
> | TROBA (Bilevel Aug  instance-wise attack)  | 65.82 | **41.39** | 53.25 | **28.05** | **40.08** | **16.88** |
>
>
> **3. [It is not clear if the adversarial example l_n^adv used in Eq. (7) is generated by a class-wise attack or instance-wise attack.]**
>
> - l_n^adv in Eq.(7) is generated by a bilevel instance-wise attack. Based on your comments, we will clarify that in the paper (blue text).

---

> ### Author Response · Authors · 2022-11-11
> **Response to your comments (2/2)**
>
> **[4. It is not clear how to update the backbone and classifier in Eq. (2) for readers. Note that the classifier is fixed in BOIL.]**
>
> - We use the MAML equation to broadly illustrate how the general meta-learning framework works in Eq. (2). We mention that BOIL fixes the classifier during inner optimization (Please see Section 3.1). However, for better clarity, we will describe the updated parameters based on the meta-learning framework.
>
> **[5. I am not sure if all the comparison methods employ the same backbone and the same epsilon of attack. Does this paper re-implement all these methods in the same framework?]**
>
> - First of all, we want to emphasize that we evaluate with **the same attack strength of \epsilon=8/255**  for all results. As we state in the caption of the tables, we employ **the same ResNet12 backbone**, and replace only the backbone of the baselines in their official codes. For a fair comparison with adversarial self-supervised models, we employ the ResNet18 backbone only for the results in Table 7. Since the checkpoints of adversarial self-supervised learning methods are implemented on the ResNet18 backbone, we additionally meta-train our encoder with ResNet18. We utilize official codes for the following baselines (i.e., MAML, AQ, RMAML, ROCL, ACL) and implement ADML, and BYORL since their codes are not officially released.
>
> **[6. It is doubtful for the experiment in Table 7. Are the results still for the 5-shot classification? Note that the splits used in CIFAR-FS and CIFAR-100 are different. Why not directly perform other models on CIFAR-FS? How many samples were used for training for TROBA and how many samples used for other models? This part should be clearly explained, otherwise the results are doubtful. ]**
>
> - Table 7 is **not** the result of the 5-shot classification. Table 7 is the result of robust full finetuning using a full dataset to train with a linear layer on top of the encoder that is widely used in self-supervised learning.
> Although adversarial self-supervised learning models are not our direct competitors, we think it is interesting to compare the quality of our visual representation trained with less dataset, weaker attack strength, and smaller iteration. Especially, TROBA is trained only for 64 classes while SSL is trained for the entire 100 classes of CIFAR-100. In addition, TROBA converges faster than SSL models where TROBA is only meta-trained for 300 epochs while SSL models are trained for 1000 epochs.  Further, while SSL models use 50,000 samples per batch, TROBA uses 20,000 samples per batch where 200 episodes (5-way 5-shot support images = 25 samples / 5-way 15-shot query set images = 75 samples per episode) are used for each batch. As shown in table 7 and table 10, TROBA shows comparable performance on CIFAR-10 and CIFAR-100, which have a similar distribution of trained distribution, and even better robust accuracy on STL-10, CARS, CUB, and AirCraft, which have a highly different distribution of trained distribution.
> Moreover, we pretrain the SSL models with CIFAR-100 and 7 steps of attack to avoid under-reproduced results since their official paper stated that SSL models should be trained on CIFAR-100 and at least 7 steps of attack. However, you commented we additionally show the experimental results on SSL models with fewer data (CIFAR-FS) and weaker attacks (3 steps) as TROBA in the following table.
> |Model| # step of attack | trained dataset |Clean     | PGD      |
> |-----------|-----------|-----------|-----------|-----------|
> | ROCL | 7 step | CIFAR-100 | 76.76 | 50.72 |
> | ROCL |3 step | CIFAR-FS | 70.14 | 20.62 |
> | TROBA | 3 step | CIFAR-FS | 74.26 | 49.38 |
> However, we want to clarify again that adversarial self-supervised learning methods are not our direct competitors.
>
> **[7. Do all the comparison methods employ the same input image size of 32 * 32?]**
> - Yes, we employ the same input image size of 32*32 for all tasks as we followed previous cross-domain meta-learning works [Ryu et al, R Volpi et al., Oh et al.]. We clearly describe the experimental details in the Supplementary.
>
> [Ryu et al.] MetaPerturb: Transferable Regularizer for Heterogeneous Tasks and Architectures, NeurIPS 2020 \
> [R Volpi et al.] Continual Adaptation of Visual Representations via Domain Randomization and Meta-learning, CVPR, 2021 \
> [Oh et al.] BOIL: TOWARDS REPRESENTATION CHANGE FOR FEW-SHOT LEARNING, ICLR 2021

---

> ### Author Response · Authors · 2022-11-25
> **Gentle reminders**
>
>
>
> Dear reviewer NDZi,\
> \
> Thanks for your constructive comments. \
> We summarize our response to help you to find our response more clearly.
>
> 1. **The main difference between a bilevel parameter augmentation and a data augmentation**\
> a. We provide an ablation experiment along with a clearer explanation.
>
> 2. **The main difference between a bilevel attack and an instance-wise attack**\
> a. We provide an ablation experiment between two attacks and explain the concept and our contribution more clearly.
>
> 3. **Unclear details of the experiment**\
> a. We revise our manuscripts based on your comments.
>
> 4. **Comparison with adversarial self-supervised learning through robust full-finetuning results**\
> a. We additionally train an **adversarial self-supervised learning model on CIFAR-FS with a weaker attack (3 steps)**, which is the same experimental setting as our TROBA.
>
> We believe we did our best to resolve your concerns. Therefore, we hope you find our response and reflect on your final review and the score. Please kindly let us know if there are any further questions or concerns. We would be more than happy to address them.
>
> We thank you again for your time and efforts in reviewing our paper.
>
>
> Best, \
> Authors

---

### Author Response · Authors · 2022-11-16
**Responses and Revision uploaded**

Dear Reviewers,

Would you please go over our responses and the revision since we can have interactions with you until this Friday (18th)? We have answered all your comments and faithfully reflected them in the revision, and provided additional experimental results that you have requested. We sincerely thank you for your time and efforts in reviewing our paper, and your insightful and constructive comments.

Thanks, Authors

---

### Author Response · Authors · 2022-11-24
**Reminder: Responses and Revision uploaded**

Dear reviewers, \
\
We politely remind you of our responses. Could you please go over our rebuttal and check our responses? We believe that we have addressed all your concerns and further your comments further strengthen our paper. We hope you reflect this in your final review and the score.
Please kindly let us know if there are any further questions or concerns. We would be more than happy to address them. We thank you again for your time and efforts in reviewing our paper. \
\
Sincerely, \
Authors.

---

### Decision · Program_Chairs · 2023-01-20

**Decision:**

Reject

**Justification For Why Not Higher Score:**

This paper is a borderline case. During the virtual discussion meeting, the reviewers have commented that the novelty of the proposed method is not enough, writing can be further improved, and results can be made more convincing via performing experiments at a larger scale. By the end, the reviewers reached the agreement to reject the paper at its current form.

**Justification For Why Not Lower Score:**

N/A

**Metareview: Summary, Strengths And Weaknesses:**

This paper develops a new algorithm for adversarially robust few-shot learning with bilevel attacks, which allows the model to learn generalizable robust representations across tasks and domains.

Initially, this paper received scores of 3566. After rebuttal, the scores have increased to 5666. The AC therefore held an online discussion meeting, where all of the 4 reviewers have joined the discussion.

On one hand, the reviewers agree that this paper is well written, and the presentation is clear. The results also look reasonably good. On the other hand, after the virtual discussion meeting, several concerns still remain. (1) The proposed method is not novel enough. Though the method contains some new design, it mostly combines different pieces of existing techniques together. (2) Results are not impressive enough. One reviewer mentioned that the results are strong, and the improvements are significant. On the other hand, other reviewers commented that the authors should probably push the experiments to a larger scale to make results more convincing.

Overall, this paper is a boderline case, and during the virtual discussion meeting, the reviewers had a thorough discussion on the pros and cons of the paper. On balance, the cons slightly outweigh the pros, and by the end, the AC decides to recommend rejection of the paper, and encouages the authors to further refine the paper and submit to a future conference.

**Summary Of Ac-Reviewer Meeting:**

The AC held a virtual discussion meeting on 12/8 10pm (GMT+8, i.e., Beijing time), that is, 12/8 9am ET/New York timezone. All the 4 reviewers have joined the discussion. During the discussion, the reviewers had a thorough discussion of the pros and cons of the paper. Overall, the reviewers agree that (1) the technical novelty of the paper is not enough, and (2) results are not impressive enough. By the end, all the 4 reviewers reached the agreement to reject the paper at its current form.